



# A daily highest air temperature estimation method and
# spatial-temporal changes analysis of high temperature in
# China from 1979 to 2018
**Ping Wang[1,2]★, Kebiao Mao[1,3]★, Fei Meng[2], Zhihao Qin[3], Shu Fang[4], Sayed M. Bateni[5],**
**Mansour Almazroui[6,7]**
[1] School of Physics and Electronic-Engineering, Ningxia University, Yinchuan 750021, China
[2] School of Surveying and Geo-Informatics, Shandong Jianzhu University, Jinan 250100, China
[3] Institute of agricultural resources and regional planning, Chinese Academy of Agricultural
Sciences, Beijing 100081, China
[4] School of Earth Sciences and Resources, China University of Geosciences, Beijing 100083, China
[5] Department of Civil and Environmental Engineering and Water Resources Research Center,
University of Hawaii at Manoa, Honolulu, HI 96822, USA
[6] Centre of Excellence for Climate Change Research/Department of Meteorology, King Abdulaziz
University, Jeddah 21589, Saudi Arabia
[7] Climatic Research Unit, School of Environmental Sciences, University of East Anglia, Norwich,
UK
Correspondence to: Kebiao Mao (maokebiao@caas.cn)
★ These authors contributed equally to this works.
**Abstract.** The daily highest air temperature ($T_{max}$) is a key parameter for global and regional high
temperature analysis, which is very difficult to be obtained in areas where there are no
meteorological observation stations. This study proposes an estimation framework for obtaining
high-precision $T_{max}$. Firstly, we build a near surface air temperature diurnal variation model to
estimate $T_{max}$ for China from 1979 to 2018 based on multi-source data. Then in order to further
improve the estimation accuracy, we divided China into six regions according to climate conditions
and topography, and established calibration models for different region. The analysis shows that the
mean absolute error (MAE) of the dataset (https://doi.org/10.5281/zenodo.5602897) is about
1.07 °C and RMSE is 1.52 °C, which improves the accuracy of the traditional method by nearly
1 °C. The spatial-temporal variations analysis of $T_{max}$ in China indicated that the annual and seasonal
mean $T_{max}$ in most areas of China showed an increasing trend. In summer and autumn, the $T_{max}$ in





northeast China increased the fastest among the six regions, which were 0.4°C/10a and 0.39°C/10a,
respectively. The number of summer days and warm days showed an increasing trend in all regions,
while the number of icing days and cold days showed a decreasing trend. The abnormal temperature
changes mainly occurred in El Niño years or La Niña years. We found that the influence of the
Indian Ocean Basin Warming (IOBW) on air temperature in China were generally greater than those
of the North Atlantic Oscillation and the NINO3.4 area sea surface temperature after making
analysis of ocean climate modal indices with air temperature. In general, this $T_{max}$ dataset and
analysis are of great significance to the study of climate change in China, especially for
environmental protection.
**Keywords**:Near surface air temperature diurnal variation model; Daily highest air temperature; High temperature;
Spatial-temporal analysis; Climate change

## 1 Introduction

In the context of global warming, the frequency of high temperature events is increasing, and high
temperature tends to increase electricity demand and energy consumption (Zhang et al., 2021;
Sathaye et al., 2013), adversely affecting human health, social economy and ecosystem (Sehra et al.,
2020; Basu, 2009; Gasparrini and Armstrong, 2011). The daily highest air temperature ($T_{max}$) is the
basic parameter for studying regional scale high-temperature events. It has a great influence on the
ozone concentration (Abdullah et al., 2017; Kleinert et al., 2021) and the start time of the plant
growth season (Yang et al., 2017). $T_{max}$ is not only an important factor for high temperature disaster
risk assessment, but also a key input parameter for crop growth models and carbon emission model.
Sustained and abnormally high $T_{max}$ will cause high temperature heat damage and adversely affect
crop growth. Therefore, it is very important to accurately obtain the temporal and spatial distribution

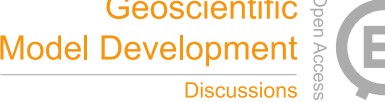

of $T_{max}$ and study the characteristics of high temperature weather. Generally, $T_{max}$ is measured on a
thermometer in a louvered box 1.5 meters above the ground in the field. Although the $T_{max}$ measured
by this method has high accuracy but not spatial continuity. Therefore, some scholars spatialize the
station based $T_{max}$ through methods such as Kriging interpolation and spline function interpolation.
However, the number of meteorological stations is limited, and stations in remote areas and areas
with complex terrain are even sparser, which makes the accuracy of $T_{max}$ obtained by interpolation
difficult to meet the requirements of regional scale research in China.
In order to obtain information about the spatial distribution of the $T_{max}$, many scholars began to
use satellite remote sensing to solve this problem. There are three commonly remote sensing
methods to estimate $T_{max}$. The first method is regression analysis, which uses the correlation
between land surface temperature (LST) and $T_{max}$ to establish a regression model to estimate $T_{max}$
(Shen and Leptoukh, 2011; Evrendilek et al., 2012; Lin et al., 2012). The second method is machine
learning, which can flexibly estimate $T_{max}$ in urban areas with complex features (Yoo et al., 2018).
The third method is to use a diurnal temperature change model to extend the instantaneous air
temperature ($T_a$) to calculate $T_{max}$, either by the Temperature-Vegetation Index (TVX) method
(Wloczyk et al., 2011; Zhu et al., 2013), the energy balance method (Sun et al., 2005; Zhu et al.,
2017), the atmospheric temperature profile extrapolation method (Fabiola and Mario, 2010), or
other methods. The above methods of estimating $T_{max}$ with LST can better reflect the spatial
distribution of $T_{max}$, but regression analysis and machine learning require sufficient and
representative samples, and the established model is not universal. TVX cannot estimate $T_a$ at night
and in sparse vegetation areas. Many parameters required by the energy balance method cannot
usually be obtained by remote sensing technology. The estimation accuracy of atmospheric

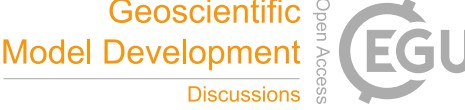

temperature profile extrapolation method is greatly affected by the accuracy of the atmospheric
temperature profile. Therefore, a new method for estimating $T_{max}$ needs to be proposed.

At present, most researches mainly use the extreme climate indices defined by the Expert Team

on Climate Change Detection and Indices (ETCCDI) to analyze the temporal and spatial distribution
characteristics of high temperature and its changing laws (Khan et al., 2018; Mcgree et al., 2019;
Poudel et al., 2020; Ruml et al., 2017; Salman et al., 2017; Wang et al., 2019; Zhang et al., 2019).
Zhou et al. (2016) analyzed the temperature indices changes in China from 1961 to 2010, and the
results indicated that the warm extremes in China exhibited an increasing trend. In addition, the
researchers analyzed the characteristics of high temperature changes in the Three River Headwaters,
Yangtze River Basin, Loess Plateau, Inner Mongolia and Songhua River Basin (Ding et al., 2018;
Guan et al., 2015; Sun et al., 2016; Tong et al., 2019; Zhong et al., 2017). In addition to analyzing
the temporal and spatial changes of high temperature events, many scholars have also studied the
influencing factors of high temperature events. Studies showed that extreme high temperature was
related to abnormal atmospheric circulation disturbances (You et al., 2011; Zhong et al., 2017) and
abnormal sea surface temperature (Li et al., 2019b; Wu et al., 2011). However, previous studies on
the cause of high temperature events usually only analyzed the correlation between atmospheric
circulation modes and the temperature indices along the time dimension, without considering the
spatial characteristics of the correlation.

From the above analysis, most of the researches mainly use the meteorological observation

temperature data interpolation to analyze local temperature changes, and almost no one constructs
continuous high-temporal resolution $T_{max}$ for high temperature analysis in China. In order to better
study regional high temperature events, this study proposes an estimation framework for obtaining



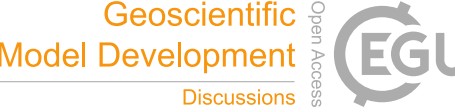

high-precision $T_{max}$. Firstly, we used multi-source data and established near surface $T_a$ diurnal
variation model to build $T_{max}$ dataset in China from 1979 to 2018. To further improve the accuracy,
we divided China into six regions according to climate conditions and topography, and established
calibration models respectively. On this basis, we further analyzed the spatial-temporal variation
characteristics of $T_{max}$ and corresponding influencing factors in China. This can provide evidence
for mitigating global climate change and reducing regional carbon emissions for environmental
protection.
**2 Study area**
In order to establish a more high-precision $T_{max}$ dataset to analyze the temporal and spatial
characteristics of high-temperature in China, we divided the country into six regions based on
topographic and climatic conditions, as shown in Fig.1. (I) The northeast region has a temperate
monsoon climate. Affected by the monsoon, $T_a$ is higher in winter in the southern part of the region,
but it is the opposite in the northern part. The topography of this area is dominated by plains, hills,
and mountains. Due to the influence of topography, the variability of $T_a$ is large in local areas. (II)
The northwestern region is dominated by a temperate continental climate (cold in winter and hot in
summer) with a large annual and daily $T_a$ range. The area is with little annual precipitation
decreasing from east to west. The topography of this area is dominated by plateau basins and rivers
are scarce. (III) North China is located in a semi-humid and humid zone in the warm temperate zone.
Precipitation is mainly concentrated in summer. This area is dominated by plains and plateaus,
bounded by Taihang Mountain, the Loess Plateau in the west, and the North China Plain in the east.
(IV) The southeast region is dominated by mountains and hills, which belongs to the warm and
humid subtropical oceanic monsoon climate zone, and the tropical monsoon climate zone. The
climate is mild, with an annual average $T_a$ of 17-21°C and an average rainfall of 1400-2000mm. (V)
The southwestern region has a subtropical monsoon climate, affected by the southeast monsoon and
southwest monsoon. It is hot and rainy in summers, and the landforms in this area are dominated by
plateaus and mountains. (VI) The Qinghai-Tibet Plateau is located in southwest China, with an
average elevation of more than 4,000 meters. The towering terrain has a great impact on the climate
in eastern and southwestern China. It has a plateau mountainous climate, with cold winters and
warm summers, with aridity and little rain throughout the year.

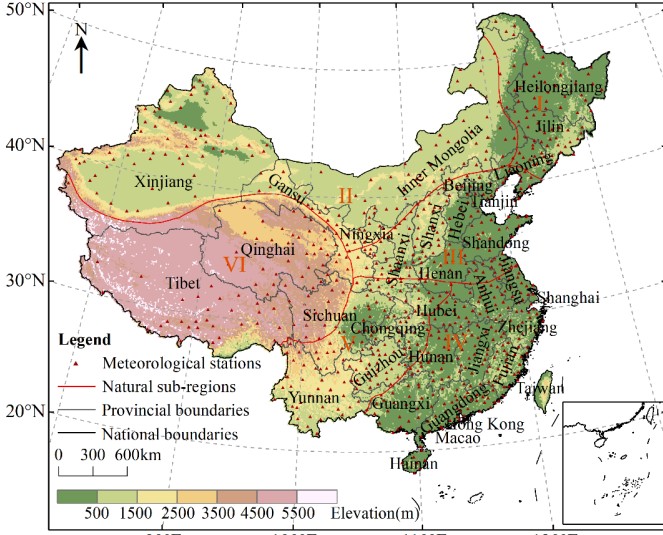


**Figure 1.** Overview of the study area.
**3 Data**
3.1 China Meteorological Forcing Dataset (CMFD)
CMFD is developed by the Hydro-meteorological Research Group of the Institute of Tibetan Plateau
Research, Chinese Academy of Sciences. The dataset can be obtained from the National Qinghai-
Tibet Plateau Science Data Center (https://data.tpdc.ac.cn/). The near surface $T_a$ data of CMFD has
a time resolution of 3h and a spatial resolution of 0.1°, and its accuracy in China is better than Global



Land Data Assimilation System (GLDAS) data (He et al., 2020). The $T_a$ data of CMFD have been
widely used in climate simulation, hydrological simulation, vegetation greenness research, and
cross-validation of new $T_a$ datasets (Luan et al., 2020; Gu et al., 2020; Wang et al., 2020). Although
this dataset has become one of the most widely used climate datasets in China, it does not provide
the $T_{max}$ value. In order to perform high temperature analysis, we need to construct a $T_{max}$ dataset.
3.2 ERA5 data
ERA5 data is the fifth generation of global climate reanalysis data produced by the European Centre
for Medium-range Weather Forecast (ECMWF) after ERA-Interim. The model version used by
ERA5 is IFS Cycle 41r2, and its spatial-temporal resolution and number of vertical layers are much
higher than the ERA-Interim data (Hoffmann et al., 2019; Urraca et al., 2018). ERA5 reanalysis
data provide a variety of meteorological elements, including atmospheric parameters, land
parameters, and ocean parameters, spanning a time range from 1950 to present. The data can be
obtained from the ECMWF ERA5 data website (https://cds.climate.copernicus.eu/). The ERA5
dataset also does not provide the $T_{max}$. This study used $T_a$ data from 1979 to 2018 to help build a
$T_{max}$ estimation model to generate $T_{max}$ value.
3.3 Meteorological station data
$T_{max}$ data from the China Surface Climatic Data Daily Dataset (V3.0) from 1979 to 2018 were used
to verify the accuracy of $T_{max}$ estimations. The hourly $T_a$ observation data from China
meteorological stations were used to determine the occurrence times of $T_{max}$ and daily lowest air
temperature ($T_{min}$). Both datasets are from CMA National Meteorological Information Center
(http://data.cma.cn/). The data were subjected to preliminary quality control and evaluation by CMA,
and all elements in the observational data are of high quality and completeness, with the validity

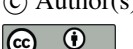



rate generally above 99%. These datasets have been widely used in Chinese climate research (Li et
al., 2019a; Tong et al., 2019). To ensure the validity of the site data, manual checks were performed
on all observed data, including extreme value tests and spatial-temporal consistency tests.
3.4 Ocean climate modal indices
The ocean occupies about 71% of the earth's surface area, which has a great impact on climate
change. After considering the distribution characteristics of China's land and sea, we analyzed the
effects of the following ocean climate modal indices on high temperature in China: Indian Ocean
Basin warming (IOBW) index, North Atlantic Oscillation (NAO) index, and NINO3.4 area sea
surface temperature (NINO3.4) index. Among them, the IOBW index comes from the National
Climate Center of CMA (http://cmdp.ncc-cma.net/cn/index.htm), and the NAO index and NINO3.4
index are from the National Oceanic and Atmospheric Administration of the United States
(https://psl.noaa.gov/data/climateindices/list/). The time range of the three indices is 1979-2018, and
the time scale is monthly.
**4 Methodology**
4.1 $T_{max}$ dataset construction
At present, the data used in the research on high temperature characteristics is mostly meteorological
station data, or grid data obtained by interpolation of station data. A limited number of stations
cannot represent the high temperature distribution at large scale. For regions where the stations are
very sparse, grid data obtained by spatial interpolation can hardly meet the accuracy requirements
of high temperature feature analysis. Although LST can be used to estimate $T_{max}$, LST has degraded
value in the presence of clouds or rainfall. Therefore, in order to obtain a $T_{max}$ dataset with high
temporal and spatial resolution, we propose a $T_{max}$ construction model that combines meteorological



station data and reanalysis data, and considers the $T_{max}$ construction under clear sky and non-clear
sky conditions. The data processing process is shown in Fig. 2, and the data construction model is
divided into two steps: $T_{max}$ estimation and $T_{max}$ correction. First, the occurrence time of $T_{max}$ and
$T_{min}$ was determined pixel by pixel. Then, $T_{max}$ was determined according to the weather state. (1)
In clear sky conditions, CMFD 3h near-surface $T_a$ data was used to construct the $T_a$ diurnal variation
model which in turn yielded $T_{max}$. (2) In non-clear sky conditions, the site and reanalysis data were
used to fill pixels. Finally, the correction model was used to correct the poor quality pixels to
generate the final $T_{max}$ dataset in China.

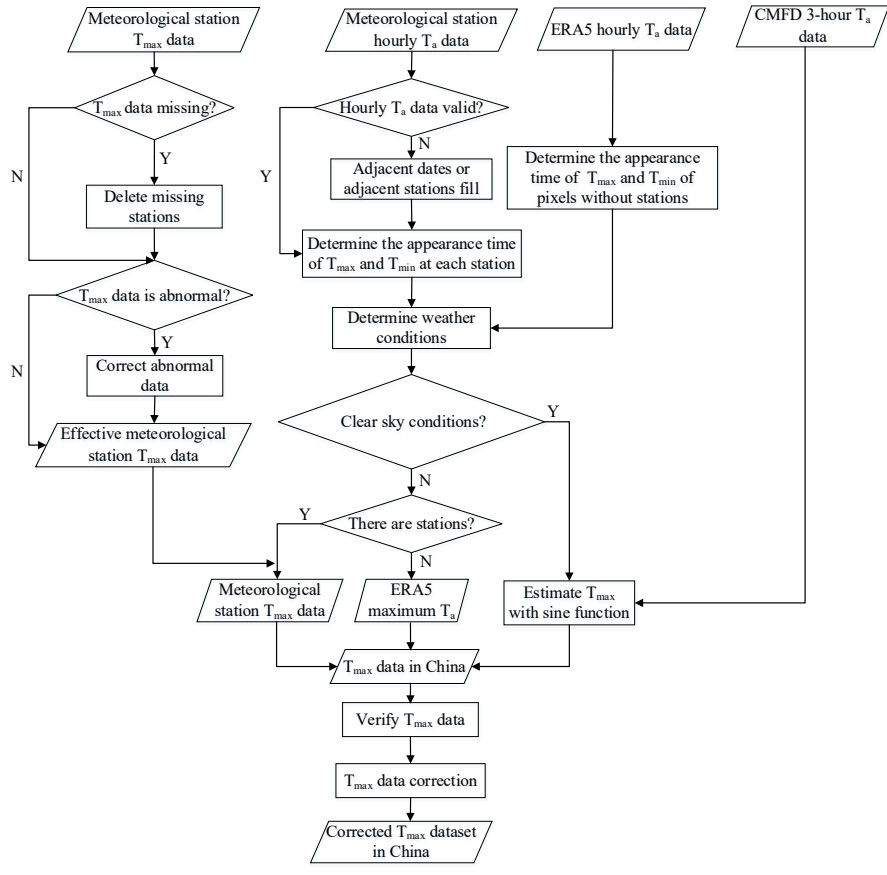


**Figure 2.** Technical roadmap for $T_{max}$ estimation.
4.1.1 $T_{max}$ estimation



The changes of $T_a$ under different weather conditions are different. The changes of $T_a$ under clear
sky conditions are relatively smooth and regular. Under non-clear sky conditions, $T_a$ changes more
drastically. In order to improve the accuracy of $T_{max}$ estimation, we determined the occurrence time
of $T_{max}$ and $T_{min}$ pixel by pixel. If there was a meteorological station at the pixel location, the analysis
could be divided into two situations. (1) If hourly $T_a$ data was valid, it was directly used to determine
the occurrence time of $T_{max}$ and $T_{min}$. (2) If there was a missing value in the hourly $T_a$ data at a
certain time, then we used the valid data from adjacent stations at the same time or adjacent time at
the same stations to fill in the missing point. If there was no meteorological station at the pixel
location, we used ERA5 hourly $T_a$ data to determine the occurrence time of $T_{max}$ and $T_{min}$.
Studies have shown that the change of $T_a$ under clear sky conditions follows a certain law: the
change curve of $T_a$ during the day is close to a sine function (Ephrath et al., 1996; Johnson and
Fitzpatrick, 1977; Parton and Logan, 1981; Zhu et al., 2013), so we used it to simulate the change
of $T_a$ during the day. The appearance time of $T_{min}$ is $t_{min}$, and the appearance time of $T_{max}$ is $t_{max}$.
According to the periodicity of the sine function, the model of the change of $T_a$ during the day is
obtained like Eq. (1).
$$T_a(t) = A sin\left(\frac{\pi(t-t_{min})}{t_{max}-t_{min}} - \frac{\pi}{2}\right) + B \tag{1}$$

Use the least square method to solve the unknowns A and B:
$$\begin{cases} \frac{\partial \delta}{\partial A} = \sum_{i=1}^{n}\left\{2*sin\left(\frac{\pi(t_i-t_{min})}{t_{max}-t_{min}} - \frac{\pi}{2}\right)*\left[A*sin\left(\frac{\pi(t_i-t_{min})}{t_{max}-t_{min}} - \frac{\pi}{2}\right) + B - T_{ai}\right]\right\} = 0 \\ \frac{\partial \delta}{\partial B} = \sum_{i=1}^{n}\left\{2*\left[A*sin\left(\frac{\pi(t_i-t_{min})}{t_{max}-t_{min}} - \frac{\pi}{2}\right) + B - T_{ai}\right]\right\} = 0 \\ \delta = \sum_{i=1}^{n}\left[A*sin\left(\frac{\pi(t_i-t_{min})}{t_{max}-t_{min}} - \frac{\pi}{2}\right) + B - T_{ai}\right]^2 \end{cases} \tag{2}$$


Here $n$ is the number of CMFD near surface $T_a$ data used to construct the $T_a$ change model in a
day. CMFD can obtain $T_a$ data 8 times a day. This study uses four daytime $T_a$ data to construct a $T_a$
variation model, so $n$ is 4. $T_{ai}$ is the near surface $T_a$ data at the $i$th time of CMFD, and $\delta$ is the sum
of squares of the difference between the model estimated $T_a$ and the near surface $T_a$ of the CMFD.
Since the change of $T_a$ under non-clear sky conditions does not conform to the sine curve change,
we divided the estimation of $T_{max}$ under non-clear sky conditions into two situations. (1) If there was
a station at the location of the pixel, the measured $T_{max}$ at the station was directly used as the





T_max of the pixel. (2) If there was no measured T_max at the pixel location, the highest value of hourly
T_a of ERA5 in a day was taken as T_max.
4.1.2 T_max correction
The validation of T_max showed some differences between the estimated T_max and the measured T_max.
In order to further improve the accuracy of T_max, the measurements taken at weather stations should
be used to correct the estimated T_max, as shown in Fig. 3. For a pixel with poor quality, if there is
station data at the location of the pixel, the low-quality pixel will be replaced with the measured
data from the station. If there is no station data at the pixel location, the data is corrected by multiple
linear regression method (Ninyerola et al., 2000; Zhao et al., 2020; Zheng et al., 2013). By
establishing the regression relationship between station T_max and estimated T_max, the residuals were
calculated according to the measured values and T_max regression predicted values, and the spatial
distribution of the residuals was obtained by the inverse distance weight (IDW) interpolation method.
Finally, the estimated T_max and the residual were added to obtain the corrected T_max. The calibration
model is like Eq. (3) and Eq. (4).
$$T_{after}(i,j) = T_{before}(i,j) + \hat{e}(i,j) \qquad (3)$$

$$\hat{e}(i,j) = T_{true}(i,j) - T_{forecast}(i,j) \qquad (4)$$

Here $i$ and $j$ are the row and column numbers of the image, $T_{after}(i,j)$ is T_max after correction,
$T_{before}(i,j)$ is T_max before correction, $\hat{e}(i,j)$ is the residual, $T_{true}(i,j)$ is the measured T_max, and
$T_{forecast}(i,j)$ is T_max predicted by the regression model.
We used the jackknife method to randomly divide the station data into calibration and
verification data (Benali et al., 2012; Zhao et al., 2020). We selected 80% of the meteorological
stations to establish the regression relationship between the measured and estimated T_max values.
The other 20% of the meteorological stations were used to verify the accuracy of the corrected data.
In order to improve data accuracy, the dataset used in the subsequent analysis of spatial-temporal
variation of high temperature was the data corrected by all stations. Due to the different topographic
and climatic characteristics of the six natural regions, the linear models of estimated T_max and
measured T_max in each region were different. In order to obtain a higher-precision correction, the six
regions were corrected separately.



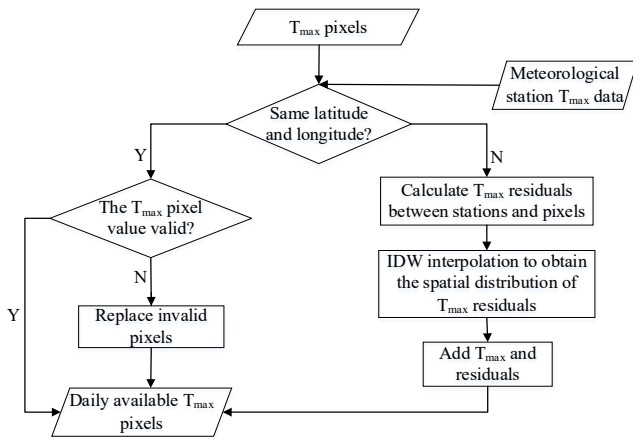

**Figure 3.** Flow chart of $T_{max}$ correction.

## 4.2 Trend analysis

### 4.2.1 Sen's slope estimation

In this study, the trends of $T_{max}$ and extreme temperature indices were calculated using Sen's slope estimation. Sen's slope estimation is a nonparametric estimation method. Even if there are some outliers in the sample, it can reliably estimate the change trend of the time series, so it is widely used in trend analysis (Sen, 1968; Zhang et al., 2017). The Eq. (5) is used to calculate the slope of each pair of data.

$$K_i = \frac{x_k - x_j}{k - j} \ (i = 1,2,\cdots,N) \tag{5}$$

Where $N = \frac{n(n-1)}{2}$, $x_k$ and $x_j$ are the time series values of the $k$th and $j$th samples respectively ($1 \leq j < k \leq n$). Arranging the $N$, $K_i$ values in ascending order, the median Sen's slope is estimated as Eq. (6).

$$Slope = \begin{cases} K_{[(N+1)/2]} & ,N \ is \ odd \\ \frac{K_{[N/2]} + K_{[(N+2)/2]}}{2} & ,N \ is \ even \end{cases} \tag{6}$$

### 4.2.2 Mann-Kendall trend test

Mann-Kendall trend test is used to test the trends of $T_{max}$ and extreme temperature indices. Mann-Kendall method does not require samples to follow a certain distribution and is not disturbed by a few outliers, and it can test the change trend of time series (Seenu and Jayakumar, 2021; Tan et al., 2019). Eq. (7) is used to calculate the statistic of the Mann-Kendall trend test.





$$S = \sum_{i=1}^{n-1} \sum_{j=i+1}^{n} \text{sgn}(x_j - x_i) \tag{7}$$

$$\text{sgn}(x_j - x_i) = \begin{cases} 1 & ,x_j - x_i > 0 \\ 0 & ,x_j - x_i = 0 \\ -1 & ,x_j - x_i < 0 \end{cases} \tag{8}$$

$$\text{Var}(S) = \frac{n(n-1)(2n+5)}{18} \tag{9}$$

Here $x_i$ and $x_j$ are the $i$th and $j$th data values of the time series, and $n$ is the length of the time series, where $n$ is 40. $\text{Var}(S)$ is the variance of $S$. The standardized statistic $Z_c$ is computed by using Eq. (10).

$$Z_c = \begin{cases} \frac{S-1}{\sqrt{Var(S)}}, & S > 0 \\ 0 & ,S = 0 \\ \frac{S+1}{\sqrt{Var(S)}}, & S < 0 \end{cases} \tag{10}$$

When $|Z_c| > Z_{1-\alpha/2}$, the change trend is considered to be significant. Here, $Z_{1-\alpha/2}$ is the standard normal variance, $\alpha$ is the significance test level, when $\alpha = 0.05$, $Z_{1-\alpha/2} = 1.96$, and when $\alpha = 0.01$, $Z_{1-\alpha/2} = 2.58$.

### 4.3 Mann-Kendall test for abrupt change analysis

Climate system change is an unstable and discontinuous change process, and one of the commonly used methods to test its change is the Mann-Kendall mutation test, which is very effective in testing the change of elements from a relatively stable state to another state (Ruml et al., 2017). For a time series $x$ with $n$ samples, Eq. (11) is used to construct an ordered sequence.

$$s_k = \sum_{i=1}^{k} r_i \quad (k = 2,3,\cdots,n) \tag{11}$$

$$r_i = \begin{cases} +1, x_i > x_j \\ 0 & ,x_i \le x_j \end{cases} (j = 1,2,\cdots,i) \tag{12}$$

$$UF_k = \frac{s_k - \text{E}(s_k)}{\sqrt{\text{Var}(s_k)}} \quad (k = 1,2,\cdots,n) \tag{13}$$

$$\text{E}(s_k) = \frac{k(k-1)}{4} \tag{14}$$

$$\text{Var}(s_k) = \frac{k(k-1)(2k+5)}{72} \tag{15}$$

Where $s_k$ is the cumulative count of the number of values at time $i$ greater than that at time $j$. $\text{E}(s_k)$ and $\text{Var}(s_k)$ are the mean and variance of the cumulative number $s_k$ respectively. $UF_k$ is a



standard normal distribution, given the significance level α, and may be obtained from the normal
distribution table. If $|UF_k| > U_\alpha$, which indicates that there is an obvious trend change in the
sequence. Reverse the time series $x$ to $x_n, x_{n-1}, \cdots, x_1$, and repeat the above process with $UB_k =$
$-UF_k(k = n, n-1, \cdots, 1)$.
### 4.4 Extreme temperature indices
ETCCDI proposed a set of extreme climate indices in the Climate Change Monitoring conference,
which became the unified standard for climate change research (Hong and Ying, 2018; Mcgree et
al., 2019; Poudel et al., 2020; Zhang et al., 2019; Zhou et al., 2016). Among them, 27 indices are
considered as core indices, which are calculated from daily $T_a$ and precipitation data and have the
characteristics of weak extremeness, low noise, and strong significance. These indices
comprehensively capture the frequency, intensity and duration of extreme climate events, and are
recommended as the core indicators for extreme climate event analysis by the STARDEX program
of the European Union (Guan et al., 2015; Ruml et al., 2017). In this study, six temperature indices
related to $T_{max}$ were used to analyze high temperature characteristics, and their definitions are shown
in Table 1.

**Table 1.** Definition of extreme temperature indices.

| Index | Name | Definition | Category | Unit |
|---|---|---|---|---|
| SU | Summer days | Annual count of days when $T_{max} > 25°C$ | Frequency | d |
| TX90p | Warm days | Annual count of days when $T_{max} > 90$th percentile | Frequency | d |
| TXn | Minimum $T_{max}$ | Annual minimum value of $T_{max}$ | Intensity | °C |
| TXx | Maximum $T_{max}$ | Annual maximum value of $T_{max}$ | Intensity | °C |
| ID | Icing days | Annual count of days when $T_{max} < 0°C$ | Frequency | d |
| TX10p | Cold days | Annual count of days when $T_{max} < 10$th percentile | Frequency | d |

### 4.5 Correlation analysis
Pearson correlation coefficient is often used to accurately measure the degree of correlation between
two variables, and its size can reflect the strength of the correlation of the variables (Cao et al., 2020;
Yan et al., 2021). For variables $x_1, x_2, \cdots, x_n$ and variables $y_1, y_2, \cdots, y_n$, the correlation coefficient
between them is calculated as Eq. (16).





$$R = \frac{n\sum_{i=1}^{n}(x_i \times y_i) - \sum_{i=1}^{n}x_i\sum_{i=1}^{n}y_i}{\sqrt{n\sum_{i=1}^{n}x_i{}^2 - (\sum_{i=1}^{n}x_i)^2}\sqrt{n\sum_{i=1}^{n}y_i{}^2 - (\sum_{i=1}^{n}y_i)^2}}$$ (16)

Here $n$ is the total length of the time series. The value of the correlation coefficient $R$ is between
-1 and 1. $R<0$ indicates a negative correlation. $R>0$ indicates a positive correlation. The closer the
absolute value of $R$ is to 1, the closer the relationship between the two elements is.
**5 Results**
5.1 Validation
In order to verify the feasibility of $T_{max}$ estimation using the $T_a$ diurnal variation model and to
analyze the accuracy of $T_{max}$ estimation in different regions, scatter plots of estimated $T_{max}$ and
measured $T_{max}$ in six natural regions (I, II, III, IV, V and VI) were drawn according to the regional
division in Fig. 1. The results are shown in Fig. 4, and the validation in each region shows that the
root mean square errors (RMSE) is between 2.38-2.94°C, and the mean absolute error (MAE) is
between 1.88-2.45°C, and the coefficient of determination ($R^2$) is between 0.95-0.99. Among them,
the accuracy in region IV is the highest, while the accuracy is the lowest in region VI. As can be
seen from Fig. 4, although most of the data is very accurate, some have some room for improvement.
Therefore, further correction is needed to improve the accuracy of the $T_{max}$ dataset.





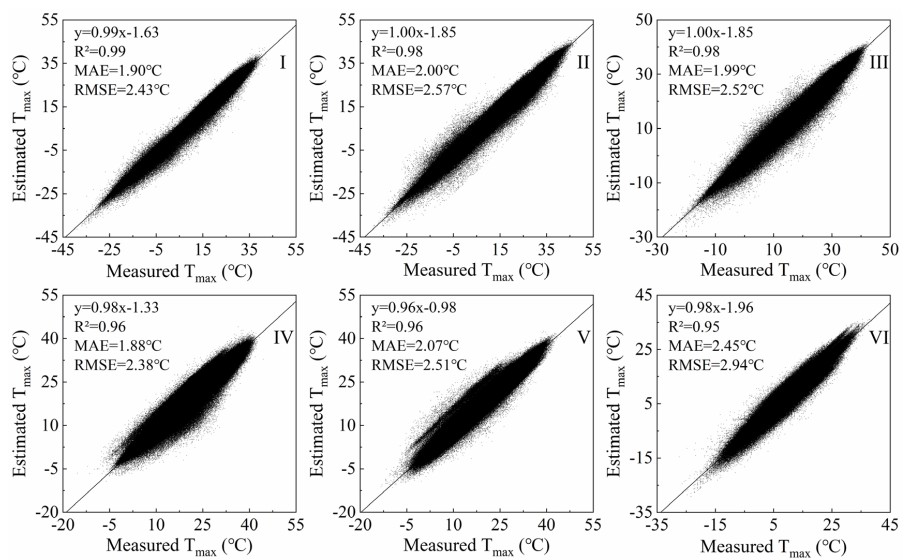

**Figure 4.** Validation of $T_{max}$ estimation results in each region.

The correction method in Sect. 4.1.2 was used to correct the $T_{max}$ estimation results of six regions

separately. The comparison between $T_{max}$ before and after correction with the measured $T_{max}$ is

shown in Fig. 5. It can be seen that $T_{max}$ corrected by the regression model is more consistent with

the measured $T_{max}$. The RMSE decreases from 2.38-2.94°C to 1.14-1.81°C, and the MAE decreases

from 1.88-2.45°C to 0.84-1.38°C, and the $R^2$ increases from 0.96-0.99 to 0.97-0.99. The accuracy

of $T_{max}$ is improved in each region after correction. The number of meteorological stations in region

I is denser, and the accuracy of $T_{max}$ after calibration is significantly improved. The RMSE reduced

from 2.32°C to 1.14°C, and the error is reduced by 51%. The number of meteorological stations in

region VI is small, and the topography is undulating and the spatial heterogeneity is large. Therefore,

the accuracy in this region is still the lowest among the six natural areas after correction. In general,

the corrected $T_{max}$ dataset has higher consistency with the measured data, and which can be applied

to research related to regional scale $T_{max}$.





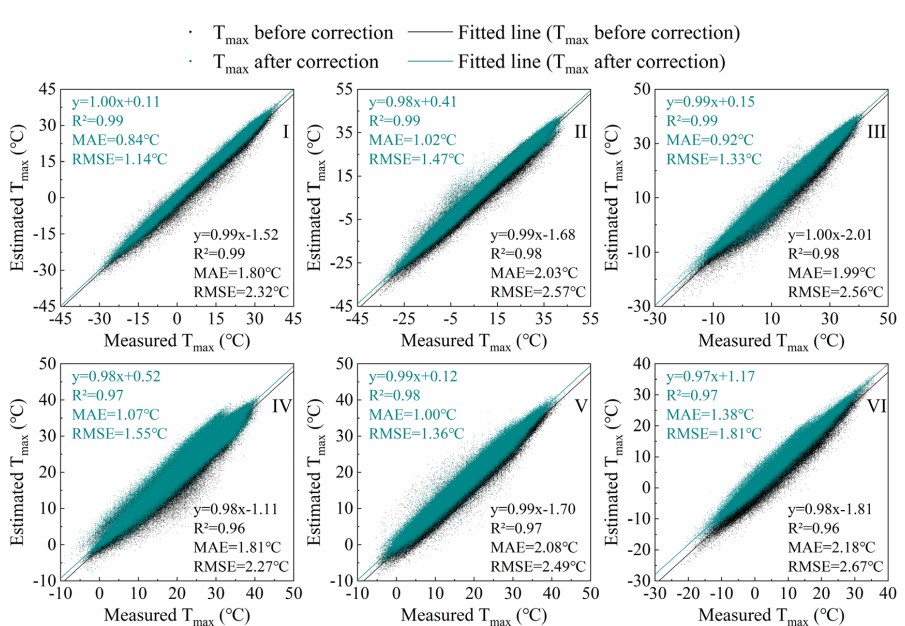

**Figure 5.** Validation of $T_{max}$ after correction.

### 5.2 Temporal and spatial changes of $T_{max}$

#### 5.2.1 Inter-annual variability

Fig. 6 shows the annual average change of $T_{max}$ in each region of China during 1979-2018. The $T_{max}$

in each region exhibited an upward trend. However, due to the different geographical locations and

the influence of atmospheric circulation in various regions, the change of $T_{max}$ was also different.

The order of the $T_{max}$ increase in each region was: V>IV>III>Whole>VI>II>I. The $T_{max}$ anomaly

ranges of region I-VI and the whole China region were -1.41-1.53, -1.54-1.16, -1.47-1.12, -1.34-

0.92, -0.97-1.33, -1.31-1.15, and -1.09-0.98℃, respectively. The $T_{max}$ variation coefficients were

0.082, 0.045, 0.036, 0.024, 0.03, 0.088 and 0.038, respectively. It can be seen that $T_{max}$ fluctuated

the most in region VI and the least in region IV. The minimum values of region I-VI and China

region appeared in 1987, 1984, 1984, 1984, 1989, 1983, and 1984, respectively which were

distributed in the 1980s. The highest values of $T_{max}$ appeared in 2007, 2007, 2017, 2007, 2013, 1999,





and 2007 respectively. Zhai et al. (2016) found that 1999, 2007, and 2013 were among the 10 years
with the highest average $T_a$ in China from 1900 to 2015. From 1998 to 2012, global surface
temperature experienced a warming hiatus (Du et al., 2019; Li et al., 2015), and $T_{max}$ in all regions
of China showed a downward trend during this period.

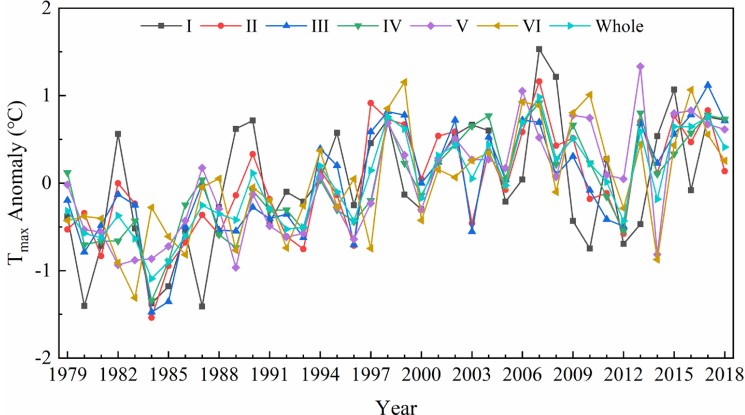

**Figure 6.** Inter-annual changes of $T_{max}$ anomalies in six regions of China during 1979-2018.

In order to understand the spatial pattern and regional differences of $T_{max}$ changes with more

detail in China, Sen's slope estimation was used to calculate the annual average $T_{max}$ change rate
from 1979 to 2018 at the pixel scale (Fig. 7a). The significance test of the $T_{max}$ change trend was
conducted by the Mann-Kendall trend test (Fig. 7b). At the same time, the average change rate of
$T_{max}$ in each region and the area percentage of significant increase and decrease of $T_{max}$ were
calculated (Table 2). The results indicated that the annual average $T_{max}$ change rate in most regions
of China (78.24% of the study area) passed the significance test with a confidence of 0.05, and 65.84%
of the pixels showed very significant changes in $T_{max}$ (P<0.01). Fig. 7a showed that the annual
average $T_{max}$ in most regions of China was on the rise, and the fastest rising rate of $T_{max}$ was in
western Yunnan. Only 8.13% of the regions in China showed a downward trend in $T_{max}$. These were
concentrated mainly in the north and south of Xinjiang, and the northwest and south of Tibet.





Among the six regions, the average $T_{max}$ change rate of region V was the largest (0.38°C/10a), and
the average $T_{max}$ change rate of region I and region II was the lowest (0.31°C/10a) (Table 2).

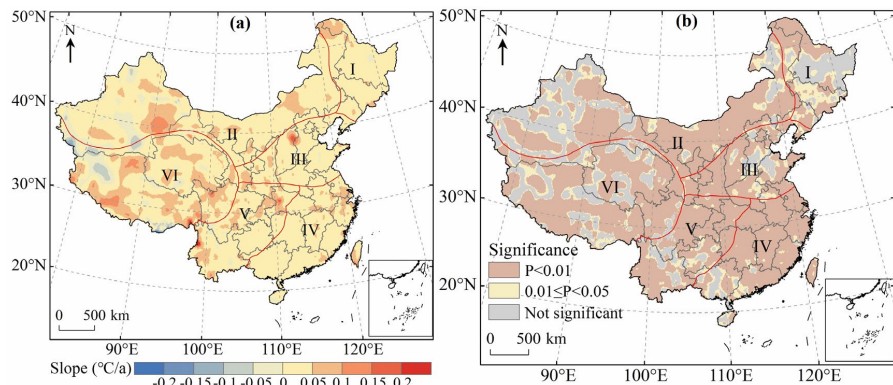

**Figure 7.** Inter-annual change rate of $T_{max}$ (a) and results of Mann-Kendall trend test (b).
**Table 2.** Statistics of $T_{max}$ change trends in various regions of China from 1979 to 2018.

| Region | I | II | III | IV | V | VI | Whole |
|---|---|---|---|---|---|---|---|
| Mean (°C/10a) | 0.31 | 0.31 | 0.33 | 0.35 | 0.38 | 0.33 | 0.33 |
| Significant upward (%) | 65.21 | 69.45 | 87.03 | 92.29 | 87.00 | 67.93 | 75.13 |
| Significant downward (%) | 0.09 | 3.14 | 0 | 0.32 | 0.75 | 7.92 | 3.11 |

5.2.2 Seasonal changes
On the basis of the annual analysis, we also analyzed the seasonal changes. The seasons are divided
according to the months (spring from March to May, summer from June to August, autumn from
September to November, and winter from December to February). We plotted the seasonal variation
curve of $T_{max}$ in China from 1979 to 2018 (Fig. 8), and some information on the trend of $T_{max}$
changes is shown in Table 3. The results indicated that $T_{max}$ in each region fluctuated the most in
winter and the least in summer. The highest $T_{max}$ in each region in spring, summer, autumn and
winter mostly occurred in 2018, 2013, 1998 and 2007, while the minimum $T_{max}$ in each region in
spring, summer, autumn and winter mostly occurred in 1988, 1993, 1981 and 1984. In 2013, $T_{max}$



of region IV-VI in summer reached the highest since 1979, mainly due to the influence of the
southwest monsoon, East Asian summer monsoon and other factors. Under the influence of El Niño,
$T_{max}$ in winter in region I, II and the whole study area was the highest in 2007. Under the influence
of La Niña, the minimum $T_{max}$ in spring and winter in most areas of China appeared in 1988 and
1984, respectively. In the same season, the variation trend of $T_{max}$ in each region was significantly
different, and some even had opposing trends. However, influenced by La Niña and the Eurasian
atmospheric circulation, $T_{max}$ in winter in each region showed a consistent decreasing trend from
2007 to 2008. As can be seen from Table 3, the regions with the fastest $T_{max}$ rise in spring, summer,
autumn and winter are III, I, I and VI respectively, and the regions with the lowest $T_{max}$ change rate
are VI, VI, III and II respectively. We found that $T_{max}$ in winter of region II exhibited a very slight
downward trend, but the sliding average of the 3-year and 5-year unit exhibited a weak upward trend.

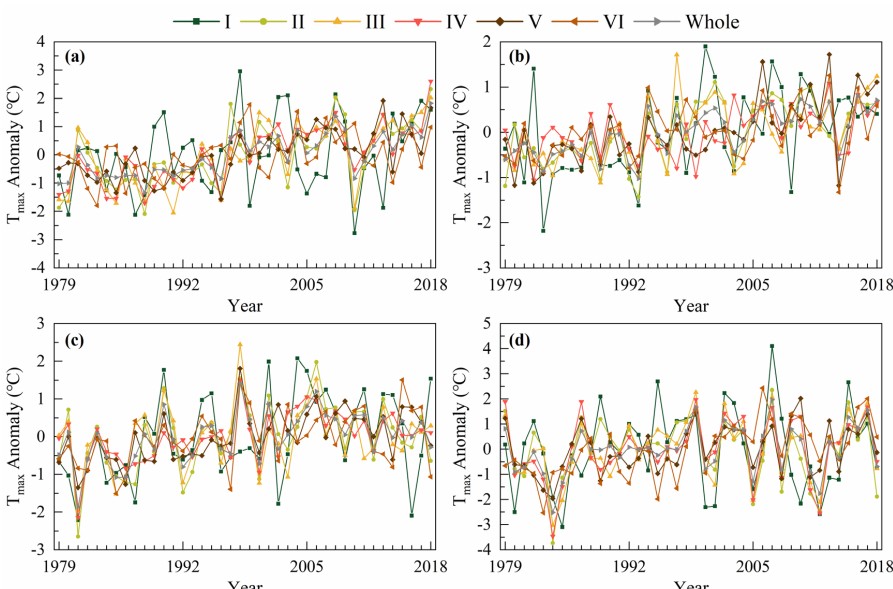

**Figure 8.** Changes of $T_{max}$ anomalies in various regions of China in spring (a), summer (b), autumn (c), winter (d)

during 1979-2018.

**Table 3.** Seasonal variation trend of $T_{max}$ in various regions of China from 1979 to 2018.



|          | I        | II       | III      | IV       | V        | VI       | Whole    |
|----------|----------|----------|----------|----------|----------|----------|----------|
| Spring   | 0.035    | 0.063**  | 0.072**  | 0.063**  | 0.051**  | 0.026*   | 0.048**  |
| Summer   | 0.040**  | 0.035**  | 0.033**  | 0.022**  | 0.039**  | 0.020*   | 0.031**  |
| Autumn   | 0.039*   | 0.024    | 0.014    | 0.025**  | 0.035**  | 0.025*   | 0.023**  |
| Winter   | 0.009    | -0.002   | 0.027    | 0.037    | 0.034*   | 0.058**  | 0.027    |

(*, ** represent the trends are significant at the level of p=0.05, p=0.01, respectively.)
In order to display the seasonal variation characteristics of $T_{max}$ in China more intuitively, we
drew the spatial distribution of the trend of $T_{max}$ and conducted a significance test (Fig. 9).
Meanwhile, we counted the percentage of significant increase and decrease of $T_{max}$ in each region
(Table 4). The results indicated that the areas with increasing $T_{max}$ were more than those with
decreasing $T_{max}$ in all seasons. From 1979 to 2018, the increasing trend of $T_{max}$ was most significant
in spring, which accounted for 92.73% of the total study area, followed by autumn and summer,
while $T_{max}$ increased the least in winter. Specifically, $T_{max}$ increased significantly in most parts of
China in spring, and the region where $T_{max}$ decreased significantly were mainly concentrated in the
region VI (Fig. 9a). In summer, $T_{max}$ in most part of China showed a significant increasing trend,
but $T_{max}$ in southern Xinjiang and northwestern Tibet exhibited a decreasing trend (Fig. 9b).
Compared with spring and summer, the area with a significant increasing trend of $T_{max}$ in autumn
was smaller, and the regions with a significant decreasing trend of $T_{max}$ were mainly distributed in
Xinjiang and Tibet (Fig. 9c). 79.02% of the regions experienced an increase in $T_{max}$ in winter, which
was significantly lower than in other seasons. A significant increasing trend of $T_{max}$ was observed
in the east of region IV, the southwest of regions V and VI, while the areas where $T_{max}$ decreased
significantly were mainly observed in Xinjiang (Fig. 9d). We also observed no significant decrease
in $T_{max}$ in regions I and III in spring, I in summer, I and IV in autumn, and III in winter (Table 4).
Further statistics showed that $T_{max}$ of the whole region III showed an upward trend in spring.

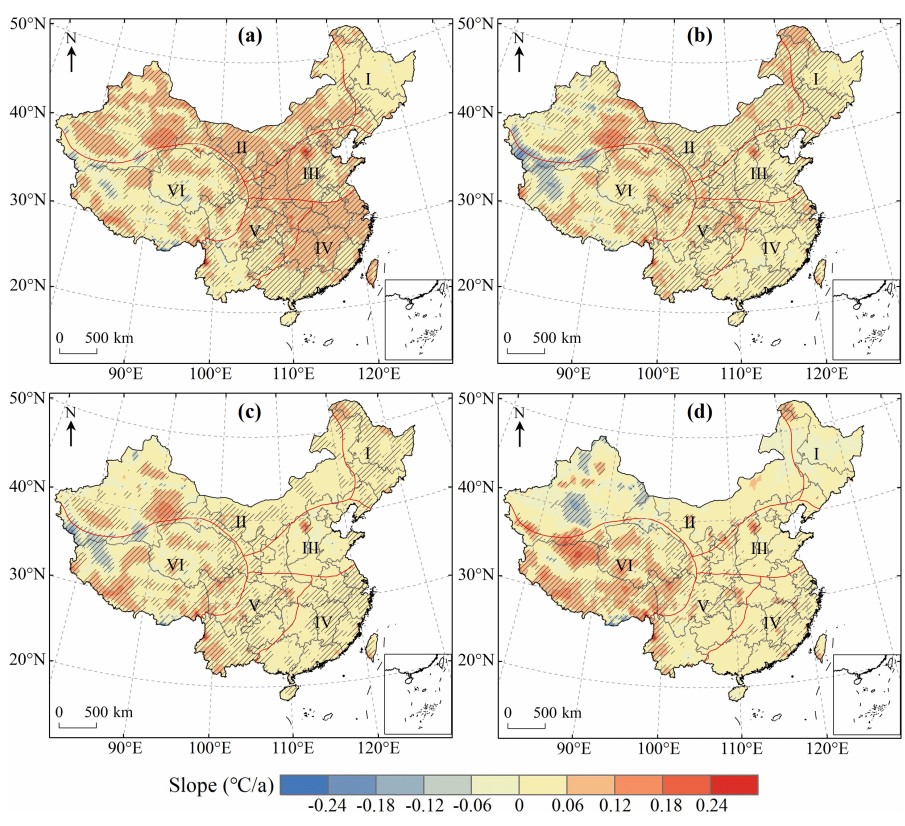

**Figure 9.** Spatial distribution of the change trend of $T_{max}$ in spring (a), summer (b), autumn (c), winter (d) over China during 1979-2018. The shaded areas indicate trends that are statistically significant at the 0.05 level.

**Table** 4. Change trend statistics of $T_{max}$ in different seasons over China from 1979 to 2018.

|  | Significant upward (%) | | | | Significant downward (%) | | | |
|---|---|---|---|---|---|---|---|---|
|  | Spring | Summer | Autumn | Winter | Spring | Summer | Autumn | Winter |
| I | 35.12 | 74.75 | 65.75 | 6.89 | 0 | 0 | 0 | 0.10 |
| II | 81.56 | 73.47 | 36.07 | 8.10 | 1.01 | 7.04 | 3.15 | 10.87 |
| III | 97.71 | 69.05 | 14.67 | 15.99 | 0 | 0.38 | 0.06 | 0 |
| IV | 96.20 | 46.80 | 57.26 | 29.47 | 0.35 | 0.68 | 0 | 0.44 |
| V | 76.48 | 75.11 | 58.56 | 31.62 | 1.24 | 1.53 | 0.06 | 0.12 |
| VI | 50.20 | 55.11 | 49.54 | 68.58 | 7.00 | 14.17 | 10.34 | 2.28 |
| Whole | 71.46 | 65.39 | 45.86 | 29.40 | 2.29 | 6.04 | 3.61 | 4.01 |

5.3 Temporal and spatial changes of extreme temperature indices

5.3.1 Change of time





We plotted the inter-annual variation of extreme temperature indices anomalies in various regions
of China from 1979 to 2018 (Fig. 10), and used Sen's slope estimation and the Mann-Kendall trend
test to calculate statistics on the trend of extreme temperature indices (Fig. 11). The results indicated
that SU, TX90p, TXn and TXx increased at a rate of 3.83d/10a, 6.57d/10a, 0.11°C/10a and
0.32°C/10a, respectively (Fig. 11). Influenced by the strong El Niño in 1997, the SU in all regions
exhibited a consistent upward trend from 1996 to 1997 (Fig. 10). The change rate of SU in all regions
passed the significance test of 0.01, indicating a significant upward trend (Fig. 11). The increasing
trend of TX90p in all regions was also very significant. The decadal average of TX90p in region
III-VI and the whole study area had an increasing trend, while the decadal average of TX90p in
region I and region II increased first and then decreased slightly. The TXn of region II showed a
weak decreasing trend, and the sliding average of the 3-year and 5-year periods also exhibited a
weak fluctuation downward trend. TXn of other regions showed an upward trend in general, and
only region VI had a significant increasing trend (P <0.05) (Fig. 11). Except for region VI, the
change rate of TXx in other regions was higher than that of TXn. The rate of change of TXx
exhibited that the upward trend of region VI was not significant, while all other regions passed the
significance test of 0.01. During 1979-2018, ID and TX10p decreased significantly at the rate of -
1.48d/10a and -3.78d /10a, respectively (P <0.01) (Fig. 11). The ID of all regions exhibited a
downward trend, with region VI and the whole study area showing the most obvious decline, passing
the significance test of 0.01 (Fig. 11). Compared with ID, TX10p decreased more sharply, and the
highest value of TX10p in all regions occurred before 1988 (Fig. 10). The above results indicate
that the frequency of high temperature events in China is on the rise, which is in line with the
expected results of global change. In addition, we also found that the occurrence time of maximum



and minimum values of SU, TXn, TXx and ID during 1979-2018 was consistent with previous
research results (Hong and Ying, 2018), which further proved the correctness of the $T_{max}$ dataset
constructed by us, indicating that built data can be used to analyze the spatial-temporal changes of
high temperature in China.

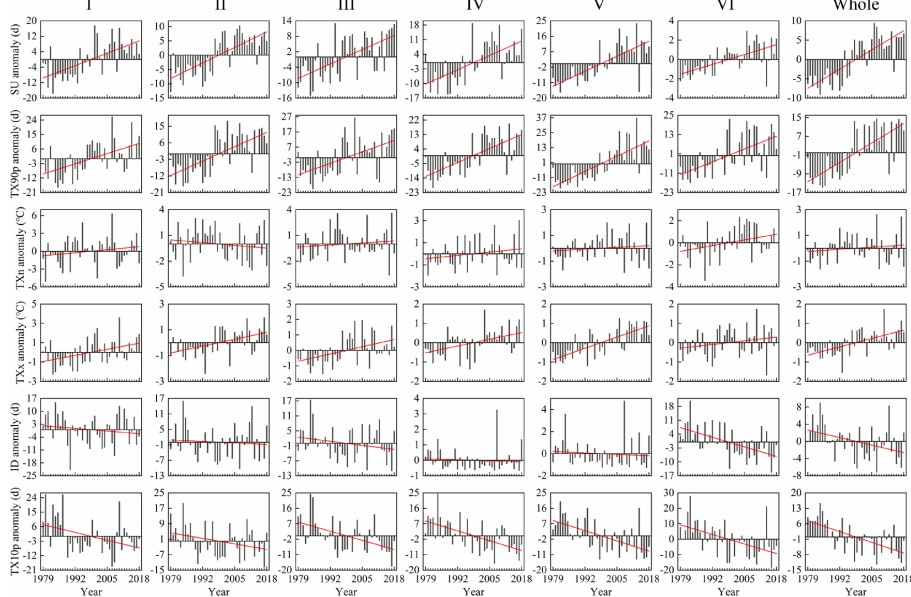


**Figure 10.** Inter-annual trend of extreme temperature indices anomalies in different regions of China during 1979-

2018.

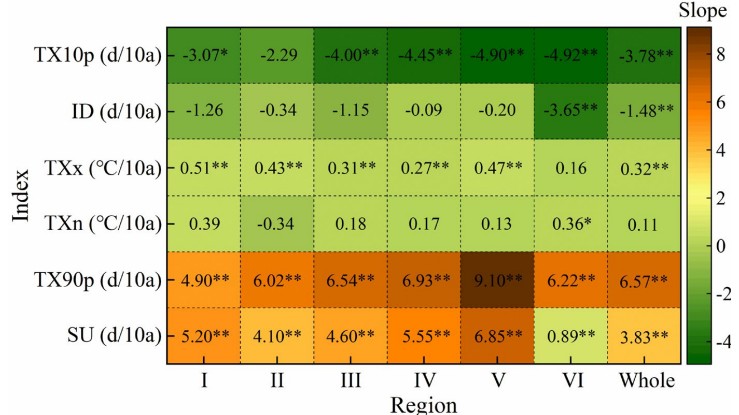


**Figure 11.** Variation trend of extreme temperature indices in different regions of China from 1979 to 2018. (*

significant at the 0.05 level, ** significant at the 0.01 level.)

In order to analyze the variation rules of extreme temperature indices in China from 1979 to
2018, the Mann-Kendall mutation test was applied to test the mutation characteristics of six extreme
temperature indices at the significance level of 0.05. The results are shown in Fig. 12. During 1979-
2018, all extreme temperature indices had abrupt changes, and 40% of the years where the abrupt
changes occurred were El Niño years, while 46.7% were La Niña years. As can be seen from the
intersection of the UF and UB curves, the SU of region III, V and VI had significant mutation in
2003, 1996 and 1990, respectively, while the other regions had no significant mutation in the whole
period of 1979-2018. TX90p in each region exhibited an overall trend of decreasing first and then
increasing. TX90p in region III was significantly mutated in 2011 and 2013, but the two mutations
did not have much influence on the trend of TX90p. The TXn of region V showed a trend of first
decreasing and then increasing, in contrast with the other regions, which all experienced a process
of increasing and decreasing many times. After mutation in region II in 2003, TXn turned from an
upward trend to a downward trend. Since the UF curve did not exceed the significance level, the
downward trend was not obvious. The TXx of region V exhibited a decreasing trend from 1979 to
1984 but was not significant. After 1984, the TXx kept rising. The UF and UB curves intersected in
1999 and were outside the significance line at the level of 0.05, indicating that the TXx of region V
had a significant mutation in 1999. The ID of the whole study area and its six sub-regions tended to
increase first and then decrease, but the upward trend was not significant. Except for region II, ID
of other regions all showed a significant downward trend. There was a long period of decline in
TX10p in all regions, and the UF value in region II was positive only in 1987. TX10p in region III-
VI and the whole study area has been in a state of significant decline since 1996, 1997, 1998, 2000
and 1993, respectively.





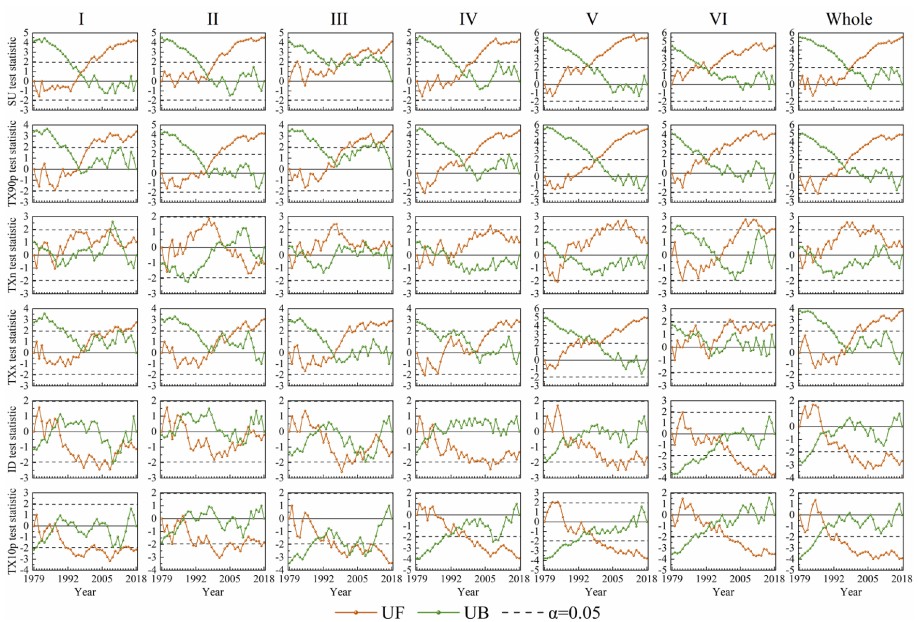


**Figure 12.** MK abrupt change detection for the extreme temperature indices in different regions of China during

1979-2018.

5.3.2 Spatial change
The spatial distribution of the extreme temperature indices trends in China during 1979-2018 is
shown in Fig. 13 (a-f), while the area percentage of the increasing and decreasing trend of extreme
temperature indices in each region is shown in Fig. 14 (a-f). For SU, TX90p, TXn and TXx, the area
with rising trend is larger than the area with declining trend. The change of SU in most regions of
China passed the significance test of 0.05, and the areas with significant increase accounted for 63.3%
of the whole study area (Fig. 14a). The regions with no significant change in SU are mainly
distributed in region VI (Fig. 13a). There are few days in a year when $T_{max}$ exceeds 25℃ in region
VI, and $T_{max}$ in some regions is even lower than 25℃ throughout the year, so the change range of
SU is small. The areas with a downward trend of TX90p were mainly distributed in southern
Xinjiang and northern Tibet (Fig. 13b). TX90p increased significantly in 75% of regions in China





(P <0.05), and the area percentage of TX90p significantly increased in region V was the largest
among the six regions (Fig. 14b). The trend of TXn change in most regions of China was not
significant, and the significant decrease was mainly concentrated in region II and region VI (Fig.
13c). While other regions were dominated by increasing trend of the TXn, 69.7% of regions in
region II showed a downward trend (Fig. 14c). For TXx, its upward trend was slightly stronger than
TXn, and the region with the highest change rate was located in western China (Fig. 13d). The
regions with significantly decreased ID were mainly distributed in region VI (Fig. 13e). 75.7% of
the regions had a declining ID, and 53% of the regions passed the significance test (Fig. 14e). As
far as TX10p is concerned, its cooling trend was much stronger than that of ID, and the areas of
significant decline were widely distributed through all regions of China (Fig. 13f). The area with a
significant decrease in region IV accounted for 75.9% of the region, which was the largest among
the six regions (Fig. 14f).

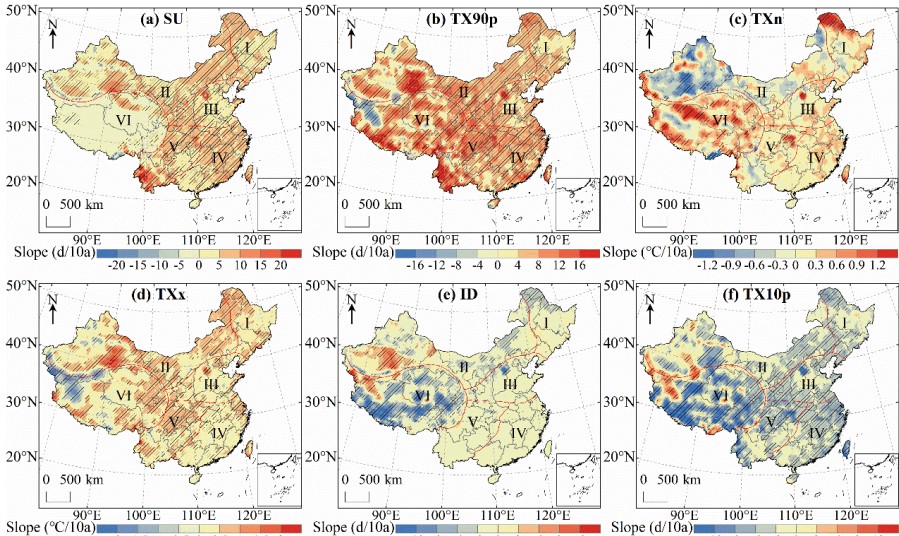


**Figure 13.** Spatial distribution of trends in extreme temperature indices over China during 1979-2018. The shaded
areas indicate trends that are statistically significant at the 0.05 level.



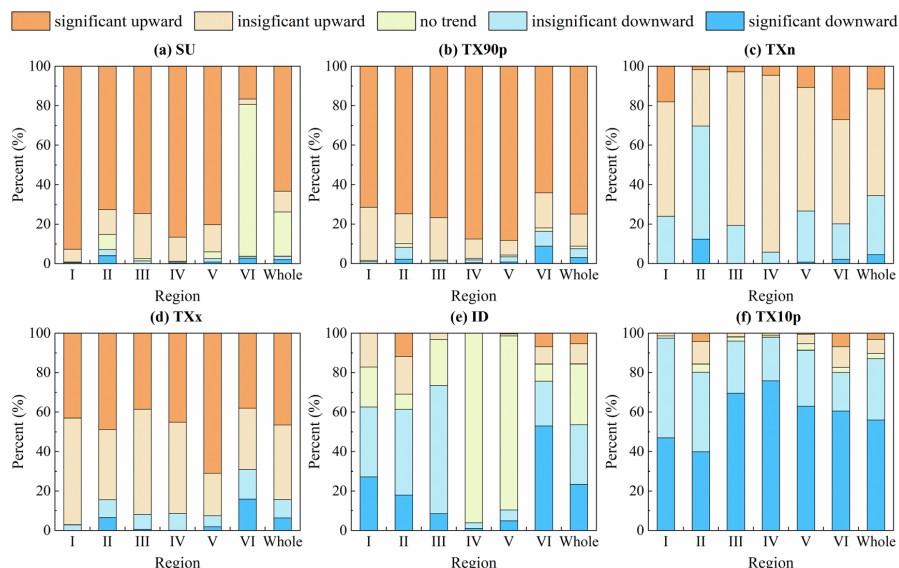


**Figure 14.** Area percentage of the trend of extreme temperature indices in different regions of China during 1979-

2018

## 6 Discussion

6.1 The influence of ocean climate modalities on $T_{max}$

The correlation between $T_{max}$ anomalies and three climate modal indices in China during 1979-2018

is shown in Fig. 15 (a-c). The results show that there is a significant positive correlation between

$T_{max}$ and IOBW in 54.18% of the regions in China, which indicates that the warming of the Indian

Ocean will lead to the warming trend of $T_{max}$ in these regions. $T_{max}$ had a moderate positive

correlation ($0.4 < R < 0.6$, $P < 0.01$) with IOBW in southern Yunnan and eastern Hainan. There were

few areas with significant negative correlation between $T_{max}$ and IOBW, only accounting for 3.28%

of regions in China, mainly concentrated in western China, which indicated that the warming Indian

Ocean will lead to the decreasing trend of $T_{max}$ in these areas (Fig. 15a). $T_{max}$ and NAO had a

significant positive correlation in northeast China, but the correlation was very weak ($R < 0.2$). The

percentage of $T_{max}$ anomaly value negatively correlated with NAO (16.55%) was higher than that



of NAO positively correlated (5.27%), mainly distributed in the west and south of region II, west of
region III, south of region IV and V, and northeast of region VI. This indicated that the positive
phase of NAO led to the decrease of $T_{max}$ in these regions (Fig. 15b). $T_{max}$ was significantly
positively correlated with NINO3.4 in southern China, central Xinjiang and southern Gansu,
indicating that El Niño events will lead to higher temperatures in these regions. In western China
and the middle part of region IV, $T_{max}$ was significantly negatively correlated with NINO3.4,
indicating that El Niño events will lead to cooling phenomena in these regions (Fig. 15c).

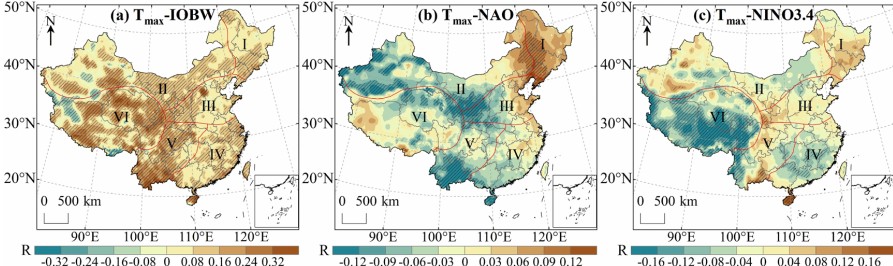


**Figure 15.** Correlation analysis between $T_{max}$ and IOBW (a), NAO (b) and NINO3.4 (c) in China during 1979-
2018. The shaded areas indicate trends that are statistically significant at the 0.05 level.
6.2 The influence of ocean climate mode on extreme temperature indices
Fig. 16 (a-f) indicates the spatial distribution of the correlation between extreme temperature indices
anomalies and IOBW in China during 1979-2018. It can be seen that SU, TX90p, TXn and TXx
over most of China are positively correlated with the IOBW. The region with significant positive
correlation between the SU and IOBW accounted for 42.67% of the whole study area, which
indicated that a warming Indian Ocean would lead to the number of days over 25°C in these regions
to increase. Significant negative correlations were found in northwest and southeast Tibet and the
mountainous regions of southern Xinjiang (Fig. 16a). The area with the largest correlation
coefficient is in the northeast of Hainan (R=0.48). The significant negative correlation between
TX90p and IOBW was mainly distributed in region VI, but the negative correlation was not strong



(|R| < 0.4) (Fig. 16b). The correlation coefficient between TXn and IOBW ranged from -0.34 to
0.34, and the regions with significant positive correlation accounted for 16.65% of the whole study
area. TXn and IOBW were significantly negatively correlated mainly in western China (Fig. 16c).
Compared with TXn, the regions with significant correlation between TXx and IOBW were more
widely distributed in China, among which the correlation coefficients in southern Yunnan and
eastern Hainan were moderately positive (0.4<R<0.6) (Fig. 16d). ID and TX10p were negatively
correlated with IOBW in most of China. The regions with significant negative correlation between
ID and IOBW were mainly distributed in region VI, and the regions with significant positive
correlation were mainly distributed in the west of region II (Fig. 16e). TX10p has a significant
negative correlation with IOBW in more areas than ID, and the significant positive correlation was
mainly located in western China (Fig. 16f).

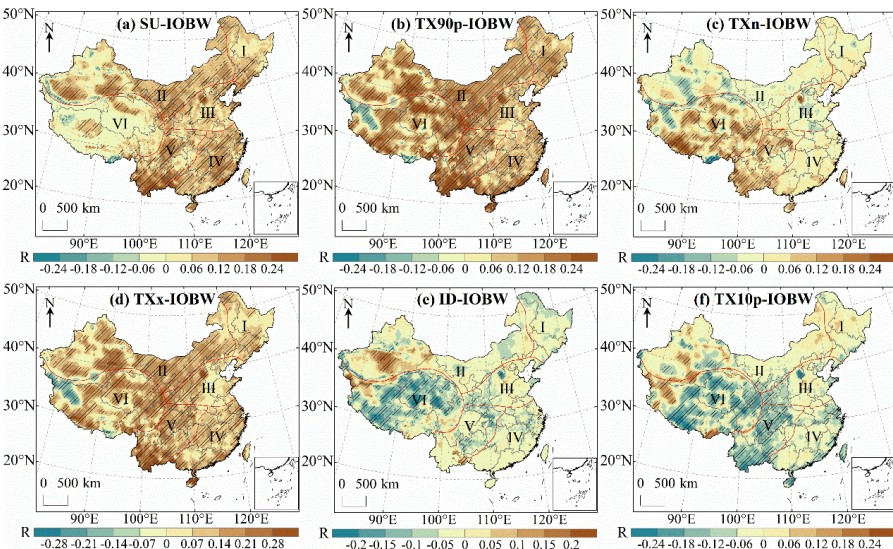

**Figure 16.** Correlation analysis between extreme temperature indices and IOBW in China during 1979-2018. The
shaded areas indicate trends that are statistically significant at the 0.05 level.





The influence of NAO on the extreme temperature indices is shown in Fig.17 (a-f). SU, TX90p,
TXx and TXn were negatively correlated with the NAO more than they were positively correlated
with NAO, indicating that the positive phase of NAO would lead to the decline of SU, TX90p, TXx
and TXn over most of China. SU and NAO had a significant positive correlation in southern
Xinjiang, western Tibet, northern Qinghai and northern Guizhou, but the correlation was very weak
(R<0.2). There was no significant correlation between SU and NAO in southern Qinghai, which
was consistent with previous observations (Ding et al., 2018). The region with the strongest negative
correlation between SU and NAO was located in Tibet (R=-0.18) (Fig. 17a). TX90p had a weak
negative correlation with NAO in eastern Xinjiang (R=-0.22, P <0.01). TX90p was significantly
positively correlated with NAO in the west and south of region VI, but the correlation was extremely
weak (Fig. 17b). Shi et al. (2019) indicated that more regions had a significant positive correlation
between TXn and NAO in China than had a significant negative correlation, which was consistent
with our results. The areas of TXn had a significant positive correlation with NAO were mainly
distributed in northeast China, while the regions with significant negative correlation were mainly
located in central Tibet, eastern Qinghai and Yunnan (Fig. 17c). The correlation coefficient between
TXx and NAO varied from -0.16 to 0.21. The regions with significant positive correlation between
TXx and NAO were mainly located in Tibet, and the region with the strongest correlation was
located in southern Tibet (Fig. 17d). The areas of ID was significantly positively correlated with
NAO accounted for 5.86% of the whole study area, and the strongest correlation was found in
western Xinjiang (R=0.23). The regions with significant negative correlation between ID and NAO
were mainly distributed in eastern and northeastern China (Fig. 17e). Sun et al. (2016) found a very
weak positive correlation between TX10p and NAO in the Loess Plateau, which was consistent with



our results. The regions with a significant negative correlation were mainly concentrated in
northeastern China (Fig. 17f).

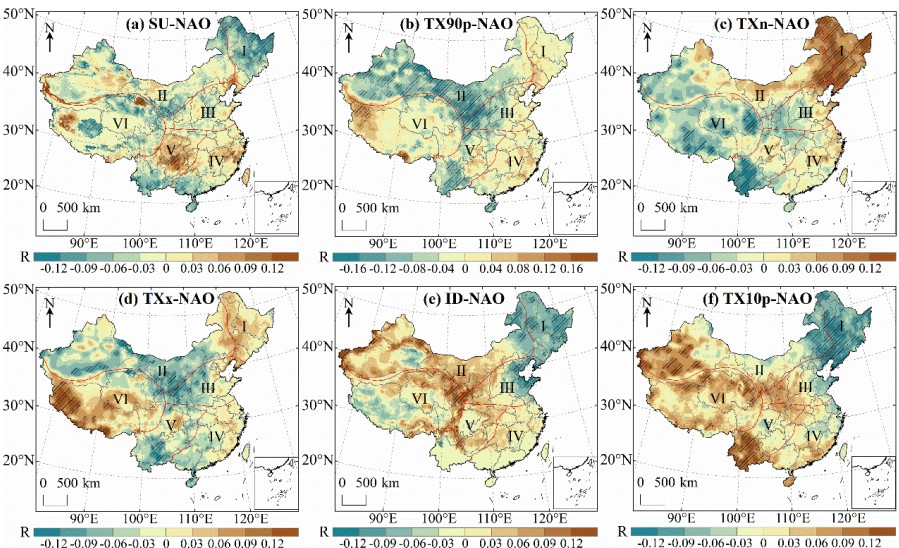

**Figure 17.** Correlation analysis between extreme temperature indices and NAO in China during 1979-2018. The

shaded areas indicate trends that are statistically significant at the 0.05 level.

Fig.18 (a-f) shows the correlation between NINO3.4 and extreme temperature indices. The
regions with significant positive correlation between SU and NINO3.4 were mainly distributed in
eastern China, indicating that the events of El Niño would lead to an upward trend of SU in these
regions. There were few regions with significant negative correlation between SU and NINO3.4,
only accounting for 1.15% of the entire research area, mainly distributed in southeast Tibet and
southwest Yunnan (Fig. 18a). The correlation coefficient between TX90p and NINO3.4 was -0.19-
0.26. The areas of TX90p had a significant negative correlation with NINO3.4 were mainly
distributed in region IV and VI (Fig. 18b). There was a significant negative correlation between
TXn and NINO3.4 in 24.59% of regions, and the region with the strongest negative correlation was
located in Tibet (R=-0.25). TXn was positively correlated with NINO3.4 in only 10.46% of regions



in China, and the region with the largest correlation coefficient was northwest Xinjiang (R=0.11)
(Fig. 18c). There was a weak positive correlation between TXx and NINO3.4 in southern
Guangdong and northern Hainan (0.2<R<0.4). The regions of TXx was significantly negatively
correlated with NINO3.4 were mainly distributed in the south of region V and region VI (Fig. 18d).
The significant negative correlation between ID and NINO3.4 was mainly concentrated in southern
China. The areas with significant positive correlation were mainly distributed in the western region
II and southern region VI, and the region with the strongest correlation was located in the western
Sichuan (R=0.31) (Fig. 18e). TX10p in most regions of regional VI was significantly affected by
NINO3.4, and the significant positive correlation area accounted for 69.31% of the whole region VI.
TX10p was significantly negatively correlated with NINO3.4 in only 0.65% of regions in China,
mainly distributed in Hainan and southern Gansu (Fig. 18f).

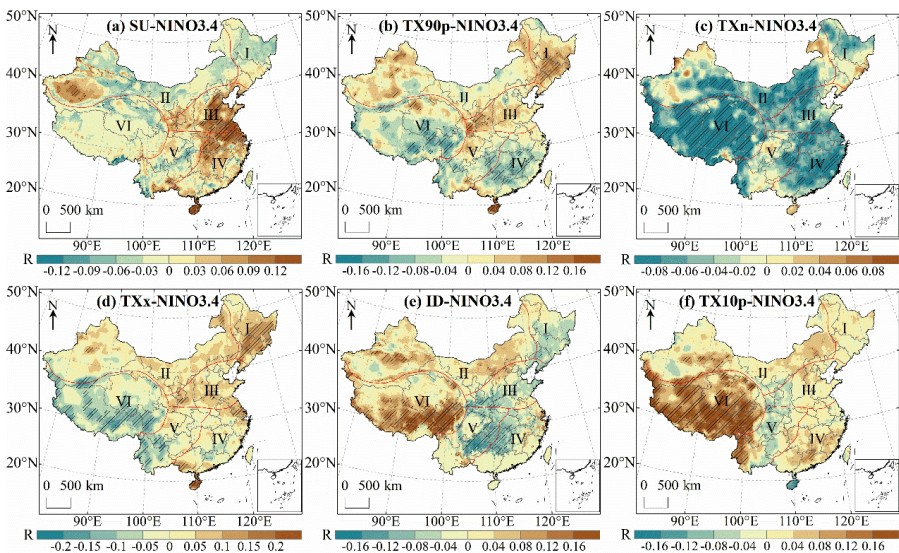

**Figure 18.** Correlation analysis between extreme temperature indices and NINO3.4 in China during 1979-2018.

The shaded areas indicate trends that are statistically significant at the 0.05 level.

## 7 Conclusions





The global temperature continues to rise and extreme weather events continue to increase. It is great
significance to study regional high temperature changes. In order to obtain the key parameters of
high temperature spatial-temporal variation analysis, this study proposed a daily $T_{max}$ estimation
frame based on the near-surface $T_a$ grid data and $T_a$ diurnal variation model to build a $T_{max}$ dataset
in China from 1979 to 2018. Validation of $T_{max}$ estimation data in six natural regions indicated that
the RMSE of each region was between 2.38-2.94°C, the MAE was between 1.88-2.45°C, and $R^2$
was between 0.95-0.99. After using the regression model to calibrate the dataset, the accuracy of
the estimated $T_{max}$ has been significantly improved. The RMSE of the $T_{max}$ after calibration reduced
to 1.14-1.81°C, and the MAE reduced to 0.84-1.38°C, and the R² increased to 0.97-0.99.

This dataset was used to study the spatial-temporal variation characteristics of $T_{max}$ and the

corresponding influencing factors in China, and to discuss the correlation between $T_{max}$, extreme
temperature indices and ocean climate modal indices. $T_{max}$ in all regions of China exhibited an
upward trend from 1979 to 2018, with the largest rise in region V and the smallest rise in region I.
In spring, $T_{max}$ in China increased significantly in most regions, and the region III is with the fastest
rising speed. In winter, $T_{max}$ in China had the least significant rise, and the region II was with the
slowest rise rate. SU, TX90p and TXx in all regions showed an upward trend. Except for region II,
TXn in other regions also exhibited an upward trend, while ID and TX10p in all regions showed a
downward trend. All extreme temperature indices had abrupt changes during 1979-2018, and most
of the abrupt changes occurred in El Niño or La Niña years. The region with the largest increase of
SU, TX90p and TXx and the region with the largest decrease of TX10p were located in the western
Yunnan. The correlation analysis between $T_{max}$ and extreme temperature indices and ocean climate
modal indices indicated that the increase of the IOBW usually led to the increase of $T_{max}$, SU, TX90p,





TXn and TXx and the decrease of ID and TX10p. NAO had the opposite relationships. In most
regions of China, $T_{max}$, SU, TX90p and TXn were negatively correlated with NINO.3.4, while TXx,
ID and TX10p were positively correlated with NINO.3.4.

The analysis of regional high temperature temporal and spatial changes shows that the

temperature changes in different regions of China are inconsistent, and the mechanism that affects
the temperature rise is different in different regions, and some regions are highly correlated with
ocean temperature changes. In order to strengthen environmental protection and control temperature
rise, and formulate reasonable carbon emission reduction measures, we need further research in the
future.

*Data availability.* CMFD is available from the National Qinghai-Tibet Plateau Science Data Center
(https://data.tpdc.ac.cn/). ERA5 data can be obtained from the ECMWF ERA5 data website
(https://cds.climate.copernicus.eu/). Meteorological station data is available by CMA National
Meteorological Information Center (http://data.cma.cn/). IOBW index can be accessed at the
National Climate Center of CMA (http://cmdp.ncc-cma.net/cn/index.htm), and NAO index and
NINO3.4 index are from the National Oceanic and Atmospheric Administration of the United States
(https://psl.noaa.gov/data/climateindices/list/).

*Author contributions.* PW and KM proposed the goals and aims of the research. FM provided
supervision and scientific guidance for the research. PW and SF built the dataset production model.
PW wrote the paper. KM, ZQ, SMB, and MA revised the final manuscript.

*Competing interests.* The authors declare no conflicts of interest.






*Acknowledgements.* The authors thank the China Meteorological Administration for providing

IOBW index and the ground measurements data, the Institute of Tibetan Plateau Research, Chinese

Academy of Sciences for providing CMFD dataset, and the NASA Earth Observing System Data

and Information System for providing the DEM data. We also thank the National Oceanic and

Atmospheric Administration of the United States for providing the ocean climate modal indices and

the ECMWF for providing the climate reanalysis data.

*Financial support.* This work is supported by the National Key Research and Development Program

of China (2019YFE0127600), National Natural Science Foundation of China (41771406), the

Fundamental Research Funds for Central Nonprofit Scientific Institution (1610132020014) and the

Open Fund of the State Key Laboratory of Remote Sensing Science (grant no. OFSLRSS201910).

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
