# Peer review of "China from 1979 to 2018"

_Geoscientific Model Development, 2021_

## Referee Comment (RC2)

**Review of the manuscript:** *"A daily highest air temperature estimation method and spatial-temporal changes analysis of high temperature in China from 1979 to 2018"* **by P. Wang et al.**

**General Comments**

The authors present a novel method for obtaining daily maximum air temperatures ($T_{max}$) over China. The proposed framework combines global reanalyses, station and satellite data, applying decision trees, regression methods and diurnal interpolation. The method is then used to construct an optimized $T_{max}$ estimation covering whole China for 40 years. As expected, the overall trend of $T_{max}$ is positive, but with regional and annual variations. Significant but small to moderate correlations to three Atmosphere and Ocean climate modal indices can be found regionally.

As the authors pointed out, their estimation method improves the accuracy of a traditional data set. Future global and regional studies on climate change will benefit from such an improvement of the China $T_{max}$ data set. Moreover, an adapted construction method may be suited for transfer to other world regions.

While the paper is written in a clearly structured way, I got a bit tired reading through all the detailed statistics in the results section. It would help the reader to get some more information on the geographical nature of the China regions in the introduction. One could do this with more visual information like climographs for selected stations of the six regions. There is also room for adding a table with basic climate information for the regions as well as the number of stations. In the discussion part, you may also elaborate on possible error sources in the trend analysis like changing data quality over time and influence of growing urban heat island effects. The Mann-Kendall test for abrupt change analysis is not well introduced and results are confusing. I'd propose to skip the respective sections (4.3 and lines 450-474 in 5.3.1). One could think of publishing this manuscript rather in a journal dedicated to climatic change, but the construction framework itself is a kind of model in geoscientific context which merits publication in GMD. Taking care of the above propositions, I generally recommend to publish the paper with minor corrections as stated out in the following.

**Specific Comments**

Lines 47-48: As pointed out by Yang et al. (2017), highest daily air temperatures have a significant influence on the end of plant growing season on the Tibetan Plateau. Evidence for an effect on the start time of the plant growth season on the Tibetan Plateau is given in Shen et al. (2016). However, these findings cannot be transferred easily to other climatic regions. Please remove association here or elaborate more detailed.

Lines 53-58: Station measurements have high accuracy for $T_{max}$, both in its value and in the time of occurrence. Spatial representativeness is of course limited. For this paper it will be of interest to know how many station observations are used for your analysis. Please refer to Figure 1 for the spatial distribution.

Line 62: Replace "land surface temperature" by "retrieved land surface temperature".

Line 86: Replace "extreme high temperature" by "extreme high temperature over China".

Line 93: "almost no one" suggests that at least a few studies (of the above cited?) have constructed a high-resolution $T_{max}$ data set for China. If so, please refer to these papers here. Otherwise, you may state that to your best knowledge no such studies exist.

Lines 107-108: What do you want to say here? It seems obvious that wintertime temperatures get higher with more southerly regions. Does "opposite" mean that the northern part of region I is not affected by monsoon?

Lines 111-112: Better say something like: "The area exhibits little annual precipitation which decreases from east to west."

Lines 129-137: Please elaborate shortly on how CMFD data has been derived.

Lines 140-141: Cite ERA5 with Hersbach et al. (2020).

Line 145: ERA5 data can be derived from the Copernicus Climate Data Store. Please change accordingly (also in the acknowledgements).

Lines 145-147: Please specify the spatial and temporal resolution of ERA5 as used in this study. You should also mention that multiple kinds of data assimilation have been performed to derive the reanalysis meteorological variables.

Lines 149-157: How many stations and observations are included in the original data set? How many of these are used in your study after performing checks and tests mentioned here? Certainly, not all stations have observational record for the whole time period. How do you deal with those stations?

Lines 175-178: You suppose here that your $T_{max}$ is constructed using station and reanalysis data only. In addition, for clear-sky condition the CMFD data is used. I argue that this data set is constructed with the help of satellite observations. Derivation of clear-sky conditions is also based on satellite retrievals?

Lines 179-180: How are the pixels mentioned here defined and which is the resolution? Is it based on satellite observations or on the era5 resolution of about 30 km? To better justify your approach, you could even give the total number of pixels and the percentage value of pixels without stations. Can you have several stations for one pixel?

Lines 217-226: I argue that the correction has been performed for all pixels, not only for pixels with poor quality (whatever this means). Please describe the correction process more clearly.

Lines 271-286: I propose to skip this section as stated above.

Lines 284-285: Rephrase sentence.

Lines 301-302: References can be omitted here.

Lines 316-317: How do you define your data as being very accurate? Please elaborate a more science-based description of Fig. 4.

Lines 372-373: How do you handle winter seasons 1978/79 and 2018/19?

Lines 387-388: Wintertime trends for region II are not significant. Omit.

Lines 440-441: Replace "… previous research results (Hong and Ying, 2018)" by "… previous research results by Hong and Ying (2018, not shown here)"

Lines 450-474: I propose to skip this section including Fig. 10.

Line 509: Replace "lead to" by "contribute to". Similarly in line 518.

Lines 510-513: Please avoid to interpret the weak negative correlation with cause and effect. This could be easily attributed to other factors which have not been examined here.

Lines 602-603: You could refer to IPCC here.

Lines 619-620: Skip sentence referring to abrupt changes.

Lines 622-626: Mention that most of the correlations shown here are small.

Line 623: Replace "led to" by "coincides with".

Lines 627-632: This section should be extended. What is the value of the new data set? How can it be used nationally and internationally? How can this study possibly interact with other international efforts for data construction?

Figure 4: Please add the total number of data points for each region.

**References:**

Hersbach, H., Bell, B., Berrisford, P., Hirahara, S., Horányi, A., Muñoz-Sabater, J., Nicolas, J., Peubey, C., Radu, R., Schepers, D., Simmons, A., Soci, C., Abdalla, S., Abellan, X., Balsamo, G., Bechtold, P., Biavati, G., Bidlot, J., Bonavita, M., De Chiara, G., Dahlgren, P., Dee, D., Diamantakis, M., Dragani, R., Flemming, J., Forbes, R., Fuentes, M., Geer, A., Haimberger, L., Healy, S., Hogan, R. J., Hólm, E., Janisková, M., Keeley, S., Laloyaux, P., Lopez, P., Lupu, C., Radnoti, G., de Rosnay, P., Rozum, I., Vamborg, F., Villaume, S., and Thépaut, J.-N.: The ERA5 global reanalysis, Q. J. Roy. Meteor. Soc., 146, 1999–2049, https://doi.org/10.1002/qj.3803, 2020.

Shen, M., Piao, S., Chen, X., An, S., Fu, Y. H., Wang, S., Cong, N., Janssens, I. A.: Strong impacts of daily minimum temperature on the green-up date and summer greenness of the Tibetan Plateau. Global Change Biology, 22(9), 3057–3066, https://doi.org/10.1111/gcb.13301, 2020.

---

## Author Comment (AC2)

Dear referee,

Thank you for your valuable comments on our manuscript and help us to improve the quality of our paper. We have carefully studied all the comments and have made revisions for manuscript. Please find our detailed responses below. Thanks again.

Sincerely,
Wang Ping and co-authors
* * *
**General comment:**
The daily highest air temperature ($T_{max}$) is a key parameter for climate change analysis. The authors proposed an estimation model to obtain high-precision $T_{max}$, and built a dataset of $T_{max}$ in China from 1979 to 2018. The authors analyzed the spatial-temporal variation characteristics of high temperature in China using the estimated $T_{max}$ data, and found that the frequency of high temperature events in most areas of China exhibited an increasing trend. The authors also showed that the temperature abrupt changes mostly occurred in El Niño years or La Niña years, indicating that China was highly vulnerable to global climate change. By analyzing the impact of ocean climate modes on temperature in China, the authors found that the warming of the Indian Ocean had a strong positive impact on warm events in most regions of China. This study is novel and significant for estimating $T_{max}$ by using multi-source data and predicting extreme weather. I suggest to accept this manuscript after minor revisions. Follows are some specific comments.
**Response: Thank you for your comprehensive review and such encouraging comments on our manuscript, all the comments have been carefully addressed. Enclosed below are our point to point responses to these comments.**

**Major comments:**
1. Section 2: How are the boundaries of regional divisions determined?
**Response: We divided China into six regions mainly based on topographic conditions (elevation), and climatic conditions (air temperature and precipitation).**

2. Section 4.1: How to solve the problem of different spatial resolutions of ERA5 data and CMFD data?
**Response: We sample the two data to the same resolution, and then use latitude and longitude as control conditions to match the different data.**

3. Section 4.4: Are the 90th percentile in TX90p and the 10th percentile in TX10p determined based on all the data in a year?
**Response: The 90th percentile in TX90p and the 10th percentile in TX10p are obtained in ascending order based on the $T_{max}$ data of each month during 1979-2018.**

4. Section 5.2: What is the significance level represented by "Significant upward" and

"Significant downward" in Table 2?

**Response: The significance level represented by "Significant upward" and "Significant downward" in Table 2 is 0.05.**

**Minor comments:**

1. L26: "region" to "regions".

**Response: Thank you for the valuable guidance, and we have made revisions.**

2. L50: "model" should be "models".

**Response: Thank you, and we have made revisions.**

3. L55: "spatialize" should be "spatialized".

**Response: Thank you for your good suggestions. We have made revisions.**

4. L77: "use" should be "used".

**Response: Thank you, and we have made revisions.**

5. L93: "use" to "used".

**Response: Thank you for the valuable comment, and we have made revisions.**

6. L94: "constructs" to "constructed".

**Response: Thank you for your guidance. We have made revisions.**

7. L178: "considers" to "consider".

**Response: Thank you, and we have made revisions.**

8. L196: I think "the missing point" should be "the missing values".

**Response: Thank you. We have made revisions.**

9. L200: Should "used it" be "used sine function"?

**Response: Thank you very much for your advice, and we have made revisions.**

10. L205: No need to express here: "Use the least square method to solve the unknowns A and B."

**Response: Thank you, and we have made revisions.**

11. L284: "may be" to "can be".

**Response: Thank you. We have modified it in the manuscript.**

12. L306: delete "the correlation coefficient".

**Response: Thank you for the valuable comment and guidance. We have made revisions.**

13. L384: "opposing" to "opposite".

**Response: Thank you, and we have made revisions.**

14. L386: "the regions with the fastest $T_{max}$ rise in spring, summer, autumn and winter are III, I, I and VI respectively" to "in spring, summer, autumn and winter, the regions with the fastest $T_{max}$ rise are III, I, I and VI respectively".
**Response: Thank you very much for your careful review, and we have made revisions.**

15. L443: "built data" to "the dataset".
**Response: Thank you, and we have made revisions.**

16. L460: "first decreasing" to "decreasing first".
**Response: Thank you for your guidance, and we have made revisions.**

17. L482: The tense of "The regions with no significant change in SU are mainly distributed in region VI" and the following sentence are not consistent.
**Response: Thank you, and we have made revisions.**

18. L605: delete "daily".
**Response: Thanks for your good suggestion. We have modified it in the manuscript.**

---

## Author Comment (AC3)

**General Comments:**

The authors present a novel method for obtaining daily maximum air temperatures ($T_{max}$) over China. The proposed framework combines global reanalyses, station and satellite data, applying decision trees, regression methods and diurnal interpolation. The method is then used to construct an optimized $T_{max}$ estimation covering whole China for 40 years. As expected, the overall trend of $T_{max}$ is positive, but with regional and annual variations. Significant but small to moderate correlations to three Atmosphere and Ocean climate modal indices can be found regionally.

As the authors pointed out, their estimation method improves the accuracy of a traditional data set. Future global and regional studies on climate change will benefit from such an improvement of the China $T_{max}$ data set. Moreover, an adapted construction method may be suited for transfer to other world regions.

While the paper is written in a clearly structured way, I got a bit tired reading through all the detailed statistics in the results section. It would help the reader to get some more information on the geographical nature of the China regions in the introduction. One could do this with more visual information like climographs for selected stations of the six regions. There is also room for adding a table with basic climate information for the regions as well as the number of stations. In the discussion part, you may also elaborate on possible error sources in the trend analysis like changing data quality over time and influence of growing urban heat island effects. The Mann-Kendall test for abrupt change analysis is not well introduced and results are confusing. I'd propose to skip the respective sections (4.3 and lines 450-474 in 5.3.1). One could think of publishing this manuscript rather in a journal dedicated to climatic change, but the construction framework itself is a kind of model in geoscientific context which merits publication in GMD. Taking care of the above propositions, I generally recommend to publish the paper with minor corrections as stated out in the following.

**Response: Thank you very much for taking the time to read our paper and giving such detailed comments, all the comments have been carefully addressed. Enclosed below are our point to point responses to these comments.**

**Specific Comments:**

Lines 47-48: As pointed out by Yang et al. (2017), highest daily air temperatures have a significant influence on the end of plant growing season on the Tibetan Plateau. Evidence for an effect on the start time of the plant growth season on the Tibetan Plateau is given in Shen et al. (2016). However, these findings cannot be transferred easily to other climatic regions. Please remove association here or elaborate more detailed.

**Response: Thanks for your guidance, we have made corresponding supplementary explanations.**

Lines 53-58: Station measurements have high accuracy for $T_{max}$, both in its value and in the time of occurrence. Spatial representativeness is of course limited. For this paper it will be of interest to know how many station observations are used for your analysis.

Please refer to Figure 1 for the spatial distribution.

**Response: Thank you, and we have added the number of meteorological ground stations in section 3.3.**

Line 62: Replace "land surface temperature" by "retrieved land surface temperature".

**Response: Thank you for the valuable guidance, and we have made revisions.**

Line 86: Replace "extreme high temperature" by "extreme high temperature over China".

**Response: Thank you for your good suggestions. We have made revisions.**

Line 93: "almost no one" suggests that at least a few studies (of the above cited?) have constructed a high-resolution $T_{max}$ data set for China. If so, please refer to these papers here. Otherwise, you may state that to your best knowledge no such studies exist.

**Response: Thank you for the valuable guidance, and we have made revisions.**

Lines 107-108: What do you want to say here? It seems obvious that wintertime temperatures get higher with more southerly regions. Does "opposite" mean that the northern part of region I is not affected by monsoon?

**Response: What I want to say is that affected by monsoon, $T_a$ in the southern part of the region is higher than that in the north in winter. "Opposite" means that $T_a$ is lower in winter in the northern part of the region I, not that the northern part of region I is not affected by monsoon.**

Lines 111-112: Better say something like: "The area exhibits little annual precipitation which decreases from east to west."

**Response: Thank you for your good suggestions. We have made revisions.**

Lines 129-137: Please elaborate shortly on how CMFD data has been derived.

**Response: ANUSPLIN software was used to interpolate the difference between GLDAS $T_a$ data and the measured $T_a$ data to obtain grid data, and then the difference grid data and the spatially downscaled GLDAS $T_a$ data were spatially added to generate CMFD $T_a$ data.**

Lines 140-141: Cite ERA5 with Hersbach et al. (2020).

**Response: Thank you for the valuable comment, and we have made revisions.**

Line 145: ERA5 data can be derived from the Copernicus Climate Data Store. Please change accordingly (also in the acknowledgements).

**Response: Thank you for your guidance. We have made revisions.**

Lines 145-147: Please specify the spatial and temporal resolution of ERA5 as used in this study. You should also mention that multiple kinds of data assimilation have been performed to derive the reanalysis meteorological variables.

**Response: Thank you, and we have made revisions. ERA5 has a temporal**

**resolution of 1h and a spatial resolution of 0.25°.**

Lines 149-157: How many stations and observations are included in the original data set? How many of these are used in your study after performing checks and tests mentioned here? Certainly, not all stations have observational record for the whole time period. How do you deal with those stations?

**Response: There are 824 stations for $T_{max}$ observation data and 2633 stations for hourly $T_a$ observation data. After performing checks and tests, we used $T_{max}$ data from 760 meteorological ground stations and hourly $T_a$ data from 2421 meteorological ground stations. If there was a missing value in the hourly $T_a$ data at a certain time, then we used the valid data from adjacent stations at the same time or adjacent time at the same stations to fill in the missing point. The processing of missing data is also reflected in Section 4.1.1.**

Lines 175-178: You suppose here that your $T_{max}$ is constructed using station and reanalysis data only. In addition, for clear-sky condition the CMFD data is used. I argue that this data set is constructed with the help of satellite observations. Derivation of clear-sky conditions is also based on satellite retrievals?

**Response: $T_{max}$ under non-clear sky conditions was constructed using station and ERA5 data, and under clear sky conditions was constructed using CMFD data. We did not construct this dataset directly from satellite observations. We used the occurrence time of $T_{max}$ and $T_{min}$ to determine clear sky conditions and non-clear sky conditions. Section 4.1.1 describes the construction of $T_{max}$ in detail.**

Lines 179-180: How are the pixels mentioned here defined and which is the resolution? Is it based on satellite observations or on the era5 resolution of about 30 km? To better justify your approach, you could even give the total number of pixels and the percentage value of pixels without stations. Can you have several stations for one pixel?

**Response: Due to the spatial resolution of our $T_{max}$ dataset is 0.1°, in order to determine the occurrence time of $T_{max}$ and $T_{min}$ for each pixel, we constructed the occurrence time images of $T_{max}$ and $T_{min}$ with spatial resolution of 0.1° based on ERA5 data and hourly $T_a$ station data. We have described this in detail in section 4.1.1. The total number of pixels is 97711. At present, there are not many meteorological stations in China, and the pixels without stations account for 97.5%. In areas where meteorological stations are very dense, there can be several stations for one pixel, but this is rare.**

Lines 217-226: I argue that the correction has been performed for all pixels, not only for pixels with poor quality (whatever this means). Please describe the correction process more clearly.

**Response: First, determine whether there is station data at the pixel location. For pixels with stations, it is further judged whether the estimated $T_{max}$ is valid by comparing the measured $T_{max}$ with the estimated $T_{max}$. If the estimated $T_{max}$ is invalid, the invalid value will be replaced with the measured data from the station.**

**If the estimated $T_{max}$ is valid, the final $T_{max}$ data is directly generated. For pixels without stations, the data is corrected by multiple linear regression method. The specific correction method is as follows: by establishing the regression relationship between the measured $T_{max}$ and the estimated $T_{max}$, the residuals were calculated according to the measured values and the corresponding $T_{max}$ regression predicted values, and the spatial distribution of the residuals was obtained by the inverse distance weight (IDW) interpolation method. Finally, the estimated $T_{max}$ and the residuals were added to obtain the corrected $T_{max}$.**

Lines 271-286: I propose to skip this section as stated above.

**Response: Mann-Kendall mutation test can help us test whether extreme temperature indices has mutation, determine the year of mutation and the change of extreme temperature indices after mutation, which is of great help to analyze the spatial-temporal change of high temperature and study China's response to global climate change.**

Lines 284-285: Rephrase sentence.

**Response: Thank you, and we have made revisions.**

Lines 301-302: References can be omitted here.

**Response: Thank you very much for your careful review, and we have made revisions.**

Lines 316-317: How do you define your data as being very accurate? Please elaborate a more science-based description of Fig. 4.

**Response: What we want to say here is that according to the calculated RMSE and MAE, the estimated $T_{max}$ accuracy of the region IV is the highest compared to other regions, not that our initial estimation of the dataset is very accurate. In addition, Figure 4 is only the preliminary estimation result of $T_{max}$, and Figure 5 shows the accuracy of the corrected dataset. Through the calculated RMSE and MAE, it can be seen that our final generated dataset has high accuracy.**

Lines 372-373: How do you handle winter seasons 1978/79 and 2018/19?

**Response: The CMFD used to make the $T_{max}$ dataset only provides data from 1979-2018. Without the support of CMFD data, it is difficult for us to complete this work. Therefore, here we only used the $T_{max}$ data from 1979 to 2018 for seasonal variation analysis.**

Lines 387-388: Wintertime trends for region II are not significant. Omit.

**Response: Thank you, and we have made revisions.**

Lines 440-441: Replace "… previous research results (Hong and Ying, 2018)" by "… previous research results by Hong and Ying (2018, not shown here)"

**Response: Thanks for your good suggestion. We have modified it in the manuscript.**

Lines 450-474: I propose to skip this section including Fig. 10.

**Response: In this section, we made an abrupt change analysis on the extreme temperature indices and got meaningful conclusions. We found that all the extreme temperature indices had abrupt change from 1979 to 2018, and the abnormal temperature changes mainly occurred in El Niño years or La Niña years. This finding further confirms that China is greatly affected by global climate change.**

Line 509: Replace "lead to" by "contribute to". Similarly in line 518.

**Response: Thank you, and we have made revisions.**

Lines 510-513: Please avoid to interpret the weak negative correlation with cause and effect. This could be easily attributed to other factors which have not been examined here.

**Response: Thanks for your good suggestion. We have modified it in the manuscript.**

Lines 602-603: You could refer to IPCC here.

**Response: Thanks for your good suggestion. We have modified it in the manuscript.**

Lines 619-620: Skip sentence referring to abrupt changes.

**Response: Through abrupt analysis of extreme temperature indices, we concluded that all extreme temperature indices had abrupt changes during 1979-2018, and most of the abrupt changes occurred in El Niño or La Niña years.**

Lines 622-626: Mention that most of the correlations shown here are small.

**Response: China is located in the eastern Eurasian continent and the western Pacific Ocean. With the influence of the unique topography of the Qinghai-Tibet Plateau, China's climate system is very complex. The temperature change in China is affected by a combination of factors, and the ocean is only one of the factors affecting the temperature change in China. Our study found that the influence of the ocean on China's temperature change is not particularly strong, and we can continue to study the driving factors that have a strong impact on China's climate change in the future.**

Line 623: Replace "led to" by "coincides with".

**Response: Thank you for your guidance, and we have made revisions.**

Lines 627-632: This section should be extended. What is the value of the new data set? How can it be used nationally and internationally? How can this study possibly interact with other international efforts for data construction?

**Response: The $T_{max}$ dataset we produced can not only be used as the input parameters of climate change models, crop growth models and carbon emission models, but also can be used to evaluate the risk of high temperature disasters,**

**which has high practical value. Currently, due to the limitation of the temporal and spatial scope of the basic data, we have only produced the dataset of China. If global station data and temperature data can be obtained in the future, we can continue to produce $T_{max}$ dataset on a global scale.**

Figure 4: Please add the total number of data points for each region.
**Response: Thanks for your good suggestion. We have modified it in the manuscript.**

---

## Referee Report (RR1)

**Comments on GMD-2021-435**

General comments:

The daily highest air temperature is an important climate variable. The research proposed a framework to estimate the daily highest air temperature. Overall, the method is effective and the results are satisfactory. However, authors can further improve the manuscript with some effort. Please see the detailed comments below.

Specific comments:

1. Abstract/Introduction: Please give the spatial resolution of the produced daily highest air temperature dataset in the Abstract and Introduction.

2. Line 26-27: I didn't find the MAE and RMSE in the Results, please supplement them in the Results. Besides, please clarify "the traditional method".

3. Introduction: Since an important part of the research is producing a new daily highest air temperature dataset, please summarize the existing daily highest air temperature datasets and their limitations.

4. Line 152: Please clarify the data assimilation you have performed.

5. Section 3: More detailed data information should be provided. Please add a table to clearly show more details about all the input data, such as title, variable, time span, time resolution, spatial resolution, reference, version, DOI/URL, date of access, preprocessing…

6. Line 236: How do you define "valid" and "poor quality"? Do you use threshold?

7. Line 238-239: Please confirm whether it is the "multiple linear regression method" or "linear regression method". Do you use any variables other than $T_{max}$?

8. Line 238-242: Is the linear regression equation established using station data of all days during 1979-2018? Is the IDW interpolation performed on each day separately? Please clarify the details.

9. Results: An overall introduction of the produced dataset is needed. Please add some paragraphs and figures to show your produced daily highest air temperature data.

10. Section 5.1: The overall performance of the dataset should be provided. Please supplement the accuracy evaluation of the dataset for the whole China.

11. Section 5.1: It will be better for data use and data improvement if more data

accuracy information is provided. Since the temperature is closely related to time, could you please supplement the accuracy evaluation of the dataset for different years and months?

12. Section 5.1: Since the highlight of the research is the production of data with the time-resolution of day, could you please supplement some evaluation to prove the dataset accuracy on the day scale?

13. Line 378: Please distinguish and regulate the use of "confidence" and "significance", it should be "with a confidence level of 95%" or "with a significance level of 0.05".

14. Table 3: Is the trend value in the table the change rate? Please clarify it in the title of the table.

15. Results: The research produced the daily data, however, the analysis are all performed at longer time scales such as season or year. Could you please supplement some analysis on the day scale to prove its usefulness and superiority?

---

## Author Response (AR2)

Dear referee,

Thank you for your valuable comments on our manuscript and help us to improve the quality of our paper. We have carefully studied all the comments and have made revisions for manuscript. Please find our detailed responses below. Thanks again.

Sincerely,
Wang Ping and co-authors
* * *
**General comments:**

The daily highest air temperature is an important climate variable. The research proposed a framework to estimate the daily highest air temperature. Overall, the method is effective and the results are satisfactory. However, authors can further improve the manuscript with some effort. Please see the detailed comments below.

**Response: Thank you very much for taking the time to read our paper and giving such detailed comments, all the comments have been carefully addressed. Enclosed below are our point to point responses to these comments.**

**Specific comments:**

1. Abstract/Introduction: Please give the spatial resolution of the produced daily highest air temperature dataset in the Abstract and Introduction.

**Response: Thanks for your guidance, we have made corresponding supplementary explanations.**

2. Line 26-27: I didn't find the MAE and RMSE in the Results, please supplement them in the Results. Besides, please clarify "the traditional method".

**Response: "the traditional method" refers to only estimating $T_{max}$ by using the near surface air temperature diurnal variation model without further correction for $T_{max}$.**

3. Introduction: Since an important part of the research is producing a new daily highest air temperature dataset, please summarize the existing daily highest air temperature datasets and their limitations.

**Response: Thank you for the valuable guidance, and we have made revisions.**

4. Line 152: Please clarify the data assimilation you have performed.

**Response: Data assimilation here refers to sampling the ERA5 data to the same spatial resolution as the CMFD data.**

5. Section 3: More detailed data information should be provided. Please add a table to clearly show more details about all the input data, such as title, variable, time span, time resolution, spatial resolution, reference, version, DOI/URL, date of access, preprocessing…

**Response: Thank you for the valuable guidance, and we have made revisions.**

6. Line 236: How do you define "valid" and "poor quality"? Do you use threshold?

**Response: If the difference between the estimated $T_{max}$ and the measured $T_{max}$ is less than 1°C, we consider the $T_{max}$ of this pixel to be valid.**

7. Line 238-239: Please confirm whether it is the "multiple linear regression method" or "linear regression method". Do you use any variables other than $T_{max}$?

**Response: Here is the "linear regression method", we only use the $T_{max}$ as a variable.**

8. Line 238-242: Is the linear regression equation established using station data of all days during 1979-2018? Is the IDW interpolation performed on each day separately? Please clarify the details.

**Response: Both the linear regression equation and IDW interpolation were performed on each day separately.**

9. Results: An overall introduction of the produced dataset is needed. Please add some paragraphs and figures to show your produced daily highest air temperature data.

**Response: Thank you for the valuable comment, and we have made revisions.**

10. Section 5.1: The overall performance of the dataset should be provided. Please supplement the accuracy evaluation of the dataset for the whole China.

**Response: Thank you for your guidance. We have made revisions.**

11. Section 5.1: It will be better for data use and data improvement if more data accuracy information is provided. Since the temperature is closely related to time, could you please supplement the accuracy evaluation of the dataset for different years and months?

**Response: Thank you, and we have made revisions.**

12. Section 5.1: Since the highlight of the research is the production of data with the time-resolution of day, could you please supplement some evaluation to prove the dataset accuracy on the day scale?

**Response: Thank you very much for your careful review, and we have made revisions.**

13. Line 378: Please distinguish and regulate the use of "confidence" and "significance", it should be "with a confidence level of 95%" or "with a significance level of 0.05".

**Response: Thank you, and we have made revisions.**

14. Table 3: Is the trend value in the table the change rate? Please clarify it in the title of the table.

**Response: The trend value in the table 3 is the change rate, and we have modified it in the manuscript.**

15. Results: The research produced the daily data, however, the analysis are all performed at longer time scales such as season or year. Could you please supplement some analysis on the day scale to prove its usefulness and superiority?

**Response: Thank you for your guidance. We have made verification and analysis of the temperature data of the daily scale. The changes in the daily -scale temperature data are mainly random, and there is no obvious laws of time and space, and no meaningful conclusions are obtained. In addition, we consider the length of the paper. Therefore, we did not make specific analysis.**